# Evolutionary dynamics of the successful expansion of pandemic *Vibrio parahaemolyticus* ST3 in Latin America

Amy Marie Campbell ●[1,2], Ronnie G. Gavilan[3,4], Michel Abanto Marin ●[5], Chao Yang[6], Chris Hauton[1], Ronny van Aerle ●[2] & Jaime Martinez-Urtaza ●[2,4] ✉

The underlying evolutionary mechanisms driving global expansions of pathogen strains are poorly understood. *Vibrio parahaemolyticus* is one of only two marine pathogens where variants have emerged in distinct climates globally. The success of a *Vibrio parahaemolyticus* clone (VpST3) in Latin America- the first spread identified outside its endemic region of tropical Asia- provided an invaluable opportunity to investigate mechanisms of VpST3 expansion into a distinct marine climate. A global collection of VpST3 isolates and novel Latin American isolates were used for evolutionary population genomics, pangenome analysis and combined with oceanic climate data. We found a VpST3 population (LatAm-VpST3) introduced in Latin America well before the emergence of this clone in India, previously considered the onset of the VpST3 epidemic. LatAm-VpST3 underwent successful adaptation to local conditions over its evolutionary divergence from Asian VpST3 isolates, to become dominant in Latin America. Selection signatures were found in genes providing resilience to the distinct marine climate. Core genome mutations and accessory gene presences that promoted survival over long dispersals or increased environmental fitness were associated with environmental conditions. These results provide novel insights into the global expansion of this successful *V. parahaemolyticus* clone into regions with different climate scenarios.

There is a compelling need to understand mechanisms that drive the rapid, global expansion of pathogen strains in the environment, to mitigate future global pandemics. *Vibrio*, a group of ubiquitous Gram-negative environmental bacteria that cause gastroenteritis, are a tangible example of a human pathogen with pandemic potential inhabiting warm coastal waters. *Vibrio* bacteria are transmitted to humans via environmental exposure routes, including recreational water use and seafood consumption, rather than human-to-human

transmission (with the exception of *Vibrio cholerae*). *Vibrio* exhibits ecological and evolutionary responses when introduced to distinct climates, displaying optimum tolerance thresholds and responses to selective pressures. Environmental conditions that are conducive to *Vibrio* survival and growth in coastal waters increase the presence of *Vibrio* in the environment which subsequently increases the disease risk for people interacting with these waters. Recent efforts have aimed to quantify possible environmental drivers of *Vibrio* prevalence, to

[1]School of Ocean and Earth Science, University of Southampton, National Oceanography Centre, Southampton, UK. [2]Centre for Environment, Fisheries and Aquaculture Science (CEFAS), Weymouth, UK. [3]Centro Nacional de Salud Pública, Instituto Nacional de Salud, Lima, Peru. [4]Department of Genetics and Microbiology, Autonomous University of Barcelona, Barcelona, Spain. [5]Genomics and Bioinformatics Unit, Scientific and Technological Bioresource Nucleus (BIOREN), Universidad de La Frontera, Temuco, Chile. [6]The Center for Microbes, Development and Health, CAS Key Laboratory of Molecular Virology and Immunology, Shanghai Institute of Immunity and Infection, Chinese Academy of Sciences, Shanghai, China. ✉e-mail: jaime.martinez.urtaza@uab.cat

facilitate sufficient monitoring, forecasting, and mitigation of resulting gastroenteritis outbreaks[1–3].

*Vibrio parahaemolyticus* specifically is the leading cause of seafood infections globally[4], with an estimated half a million cases in 2020[5]. Before the 1990s, *Vibrio* infections were considered rare, exotic outcomes of foreign travel to tropical Asian waters, apart from particular *V. cholerae* strains that underwent global expansion through pandemic waves[6]. Now, *V. parahaemolyticus* is globally prevalent, with rising infection rates, particularly in increasingly higher latitudes. Climate trends have been suggested as responsible- the warming of 70% of the world's coastline[7] is increasing the spatial ranges of optimum suitability for *Vibrio* bacteria, and extending the 'Vibrio season' (characteristically summer months) as warmer temperatures encroach into cooler months[8].

*V. parahaemolyticus* is genomically diverse, with a background of numerous genotypes from which effective clonal complexes emerge, characteristic of a classic epidemic bacterial population structure[9]. One of these effective clonal types, Sequence Type 3 (VpST3), was reported to emerge in India in 1996[10] and underwent global expansion into a breadth of distinct marine climates, supplanting local serotypes to become one of the most dominant *V. parahaemolyticus* strains currently in circulation. This widespread expansion and transoceanic dispersal had not yet been seen for a *V. parahaemolyticus* serotype, leading to VpST3 being referred to as a 'pandemic clone'. Within months of its initial detection in Asian countries, VpST3 was reported for the first time outside of this endemic region, in the vastly distinct climate of Latin America. This discovery of a VpST3 isolate in February 1996 in Trujillo[11], a city in coastal north-western Peru, was considered an epidemic expansion from the emergence in India. This was followed by a local dispersal of vibriosis outbreaks attributed to VpST3, with outbreaks reported progressively from north to south coastal regions in the country- radiation that was notably spatiotemporally correlated with the movement of El Niño waters[12,13]. By 1997, the clonal type had reached Chile, firstly with outbreaks reported in Antofagasta in Northern Chile from November 1997 to March 1998, and again in Puerto Montt, Chile in 2004[14]. This provided the first evidence for *V. parahaemolyticus* as a second marine pathogen species that is capable of worldwide expansions into distinct climates- notably the cooler, more saline and acidic marine environment on Latin America's west coast. There are few examples of pandemic expansion of environmental bacteria, with sufficient biological data to explore possible hypotheses. VpST3, and its dynamics in Latin America, therefore provides a unique, almost singular, opportunity to investigate this pandemic expansion process and settlement of strains into new areas, to understand the wider dynamics of pandemic expansions.

The mechanisms behind the introduction and establishment of VpST3 in Latin America are currently unknown. Both ecological and evolutionary components are involved in the process of the introduction, survival and establishment of environmental bacteria. *Vibrio* ecology, and resulting disease dynamics, are intrinsically linked to environmental factors[5]. Notably, water temperature and salinity determine *Vibrio* survival and trigger rapid population growth in optimum conditions. It is important to explore distinct climate conditions facilitating the successful expansion and establishment of pathogen populations in diverse locations. From an evolutionary perspective, delineating the structure of clonal VpST3 populations circulating around the same time will elucidate the characteristics of those that were successfully introduced into Latin America, and able to establish themselves. These components interact, for example, a clonal type might thrive in a distinct climate because it is better suited to the new conditions, or because it has a higher plasticity and is able to adapt to them. This therefore requires an eco-evolutionary approach to understand the pathogen responses and future global expansions amidst a changing climate.

This study aimed to uncover the dynamics of the expansion of VpST3 into Latin America using genomic data from clinical and environmental samples as a proxy reflecting the dynamics of *Vibrio* in the environment, to reconstruct the evolutionary history and dynamics of this clonal type. A global collection of 280 publicly-available VpST3 genomes, and 32 novel genomic samples from Latin America, were utilised to reconstruct the population structure of VpST3 globally and quantify the genetic differentiation between samples found in Latin America to those dominant in Asia, and within sub-populations themselves. Among a range of complex interactions, we assessed the potential role of the distinct Latin American marine environmental context in this process, contributing to the successful introduction and establishment of this strain within Latin America's distinct climate.

## Results
Our analysis of a global collection of 312 clinical and environmental isolates (see Table S1) revealed dynamics of VpST3 expansion (Fig. 1). The addition of 32 novel strains (Table 1, highlighted in Table S1) provided a more geographically-balanced outlook to previous studies, particularly due to the increased representation of Latin American strains within the collection (22.8%), amidst those from Asia (52.2%) and North America (25%). A maximum-likelihood phylogeny highlights Asia as the most diverse geographical region, with Asian samples distributed throughout and providing the source for many sub-lineages, while more distinct groups of the samples from North America and Latin America highlight independent establishment of sub-lineages.

### Global dynamics of VpST3
As our genomic collection had a strong temporal association with root-to-tip divergence (correlation coefficient of 0.68, see Figure S1), we situated the evolution of VpST3 in space and time by reconstructing a Bayesian, rooted, time-sensitive phylogeny (Fig. 2a). Our analysis identified differentiation within the clone itself; within *V. parahaemolyticus* as a species, VpST3 is a successful clonal group, but we found 3 sub-populations existing within the VpST3 clone itself; defined here as LatAm-VpST3, the Asian-dominant Cluster and the Modern Cluster. Spatial and temporal inferences of the time and location of most-recent common ancestors (MRCA) estimated the origin of VpST3 (where it diverges from existing similar *V. parahaemolyticus* populations, represented by the pre-pandemic isolate from Japan in 1980) around 1943 (confidence interval of 1889–1976), in Asia (91% confidence). The VpST3 collection then branches off into two groups, around 1963–4 (95% highest posterior density of 1940–1982), into LatAm-VpST3 (posterior support of 0.837) and the Asian-dominant Cluster (posterior support of 0.919), and the latter then diverges again, around 1995, with the Modern Cluster emerging (posterior support of 1). Geography (the continent or country in which the isolate was reported) was found to have a small, but significant, phylogenetic signal within our collection (K < 0.25, *p* < 0.01) (see Table S2).

### A Latin American sub-population
Unlike the Asian-dominant Cluster, LatAm-VpST3 predominantly contained isolates from Latin America (Fig. 2b), initially particularly from Peru, including the earliest recovered isolates in Latin America. LatAm-VpST3 contained 43 isolates recovered from 1997 to 2015, 72% of which were from Latin America, which represents over half of all the Latin American isolates in the collection. LatAm-VpST3 was estimated to diverge from the rest of the VpST3 collection around 1963-4 (95% highest posterior density of 1940–1982), arriving in Latin America, from a notably long branch, with an MRCA in Latin American estimated around 1984 (95% highest posterior density of 1978–1994), a posterior probability of 0.837, and a 0.869 probability the location of this branch was Latin America.

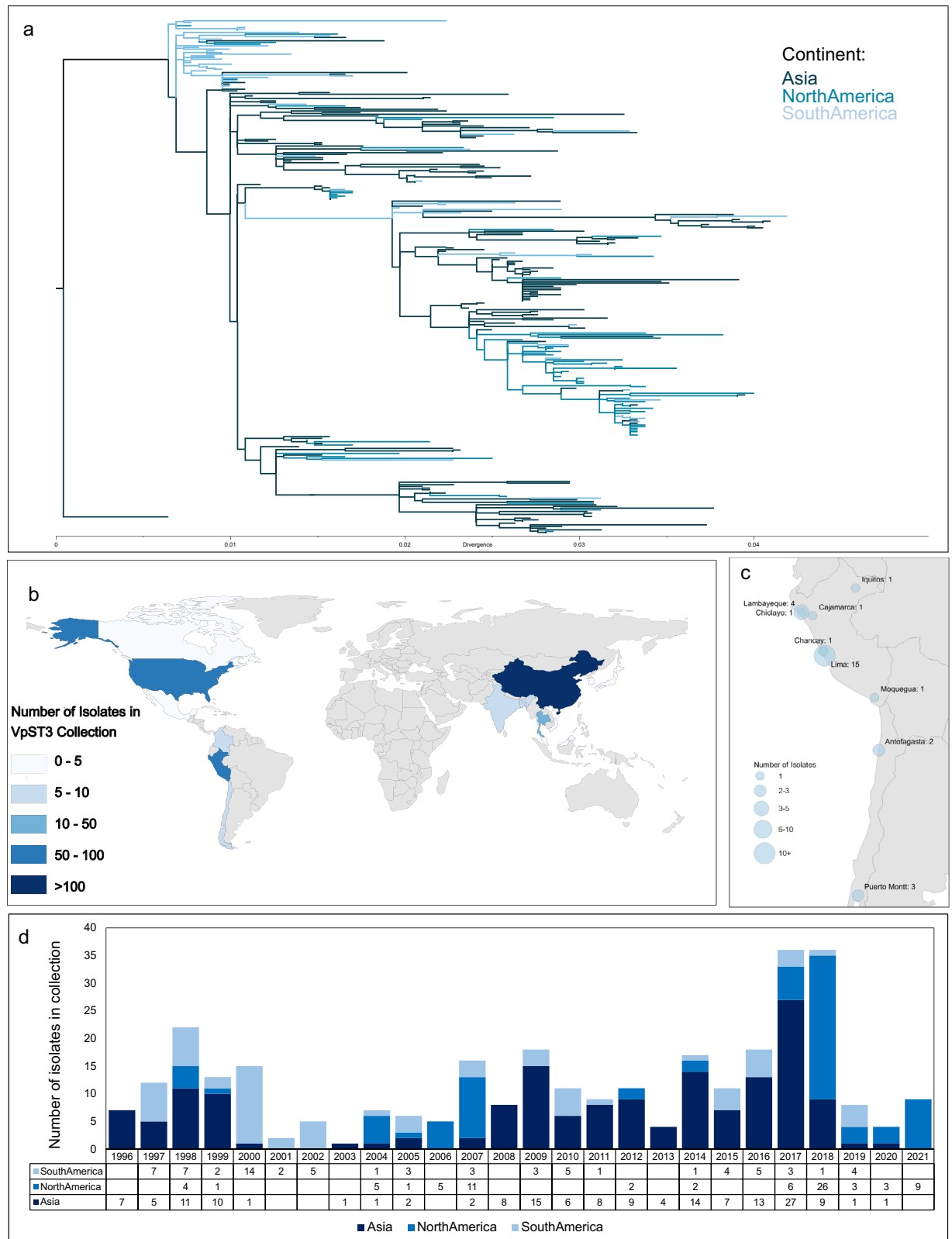

**Fig. 1 | Global collection of VpST3 genomes with the addition of novel Latin American strains. a** Maximum likelihood phylogeny using non-recombining alignment of VpST3 collection, rooted to pre-pandemic sample (Japan, 1980, see Table S1), coloured by continent of isolate origin; **b** Global distribution of VpST3 collection choropleth map displaying number of isolates analysed from each country; **c** Regional distribution of novel samples from the National Centre for Public Health in Peru (excluding two Peruvian samples without regional metadata); **d** Temporal distribution of samples in the VpST3 collection per continental origin.

**Table 1 | Isolation metadata and quality control for 32 novel VpST3 strains**

| Sample Names | Sequencing technique | Sequencing output (bp) | Mean Q30 to Base R1 | Mean Q30 To Base R2 | Total genome size (bp) | n° contigs | N50 (bp) | n° Contigs >1000 bp | Coverage |
|---|---|---|---|---|---|---|---|---|---|
| A1078098_1998-1-01_SouthAmerica_Peru | Illumina | 3037235400 | 150 | 150 | 5,098,463 | 53 | 461,842 | 40 | 588X |
| A11A1198_1998-1-01_SouthAmerica_Peru_Lima | Illumina | 252726900 | 150 | 150 | 5,217,388 | 58 | 428,132 | 43 | 49X |
| A12A1298_1998-1-01_SouthAmerica_Peru_Lima | Illumina | 336173100 | 150 | 150 | 5,095,988 | 57 | 334,356 | 42 | 65X |
| A3A397_1997-1-01_SouthAmerica_Peru_Chancay | Illumina | 859855200 | 150 | 150 | 5,193,685 | 56 | 312,677 | 42 | 166X |
| A4A497_1997-1-01_SouthAmerica_Peru_Cajamarca | Illumina | 739590600 | 150 | 150 | 5,149,414 | 60 | 334,356 | 45 | 143X |
| A5A597_1997-1-01_SouthAmerica_Peru_Lambayeque | Illumina | 477434400 | 150 | 150 | 5,104,770 | 64 | 289,565 | 48 | 92X |
| A6A697_1997-1-01_SouthAmerica_Peru_Lima | Illumina | 138352200 | 150 | 150 | 5,158,059 | 70 | 255,583 | 54 | 27X |
| A8A897_1997-1-01_SouthAmerica_Peru_Lambayeque | Illumina | 482647800 | 150 | 150 | 5,184,291 | 62 | 340,201 | 48 | 93X |
| A9A997_1997-1-01_SouthAmerica_Peru_Moquegua | Illumina | 528629700 | 150 | 150 | 5,178,062 | 71 | 288,921 | 56 | 102X |
| B10B10O_2000-1-01_SouthAmerica_Peru_Lima | Illumina | 602956800 | 150 | 150 | 5,104,271 | 61 | 289,565 | 45 | 117X |
| B11B11O_2000-1-01_SouthAmerica_Peru_Lima | Illumina | 637577100 | 150 | 150 | 5,098,843 | 59 | 334,356 | 45 | 123X |
| B12B120_2000-1-01_SouthAmerica_Peru_Lima | Illumina | 445713900 | 150 | 150 | 5,035,633 | 54 | 388,418 | 44 | 86X |
| B3B399_1999-1-01_SouthAmerica_Peru_Lima | Illumina | 714816600 | 150 | 150 | 5,104,628 | 57 | 347,413 | 43 | 138X |
| B4B499_1999-1-01_SouthAmerica_Peru_Lima | Illumina | 611731500 | 150 | 150 | 5,180,717 | 57 | 340,345 | 44 | 118X |
| B8B80_2000-1-01_SouthAmerica_Peru_Lambayeque | Illumina | 102436800 | 150 | 150 | 5,071,641 | 58 | 304,117 | 54 | 20X |
| B9B90_2000-1-01_SouthAmerica_Peru_Lima | Illumina | 746250300 | 150 | 150 | 5,237,354 | 58 | 427,280 | 44 | 144X |
| C10C101_2001-1-01_SouthAmerica_Peru_Iquitos | Illumina | 428489700 | 150 | 150 | 5,188,427 | 56 | 427,036 | 41 | 83X |
| C11C112_2002-1-01_SouthAmerica_Peru_Lima | Illumina | 323965800 | 150 | 150 | 5,172,805 | 89 | 198,175 | 69 | 63X |
| C12C122_2002-1-01_SouthAmerica_Peru_Lima | Illumina | 168009300 | 150 | 150 | 5,183,728 | 57 | 392,233 | 43 | 33X |
| C1C1O_2000-1-01_SouthAmerica_Peru_Huaral | Illumina | 1289147100 | 150 | 150 | 5,102,914 | 53 | 312,835 | 39 | 250X |
| C2C20_2000-1-01_SouthAmerica_Peru_Lima | Illumina | 817404900 | 150 | 150 | 5,121,698 | 56 | 347,630 | 43 | 158X |
| C4C41_2001-1-01_SouthAmerica_Peru_Lambayeque | Illumina | 548107200 | 150 | 150 | 5,098,297 | 59 | 334,356 | 45 | 106X |
| D1D12_2002-1-01_SouthAmerica_Peru_Lima | Illumina | 733331100 | 150 | 150 | 5,102,272 | 60 | 312,811 | 45 | 142X |
| D2D22_2002-1-01_SouthAmerica_Peru_Chiclayo | Illumina | 398510100 | 150 | 150 | 5,177,250 | 61 | 336,355 | 48 | 77X |
| D3D32_2002-1-01_SouthAmerica_Peru_Lima | Illumina | 953269800 | 150 | 150 | 5,101,837 | 58 | 334,356 | 45 | 185X |
| F1F1_2007-1-01_SouthAmerica_Chile_PuertoMontt | Illumina | 504072000 | 150 | 150 | 5,091,206 | 73 | 267,905 | 56 | 98X |
| F2F2_1998-1-01_SouthAmerica_Chile_Antofagasta | Illumina | 563799000 | 150 | 150 | 5,032,200 | 61 | 268,333 | 47 | 109X |
| F4F47_2007-1-01_SouthAmerica_Peru_ | Illumina | 607019100 | 150 | 150 | 5,102,814 | 54 | 433,108 | 40 | 118X |
| F7F7_2007-1-01_SouthAmerica_Chile_PuertoMontt | Illumina | 348476700 | 150 | 150 | 5,096,747 | 56 | 312,654 | 41 | 67X |
| G1G1_2005-1-01_SouthAmerica_Chile_PuertoMontt | Illumina | 703807800 | 150 | 150 | 5,103,511 | 55 | 461,882 | 39 | 136X |
| G2G2_1998-1-01_SouthAmerica_Chile_Antofagasta | Illumina | 186309600 | 150 | 150 | 5,100,712 | 53 | 401,005 | 37 | 36X |
| G3G3_1998-1-01_SouthAmerica_Peru_Lima | Illumina | 713747400 | 150 | 150 | 5,100,608 | 70 | 288,741 | 50 | 138X |

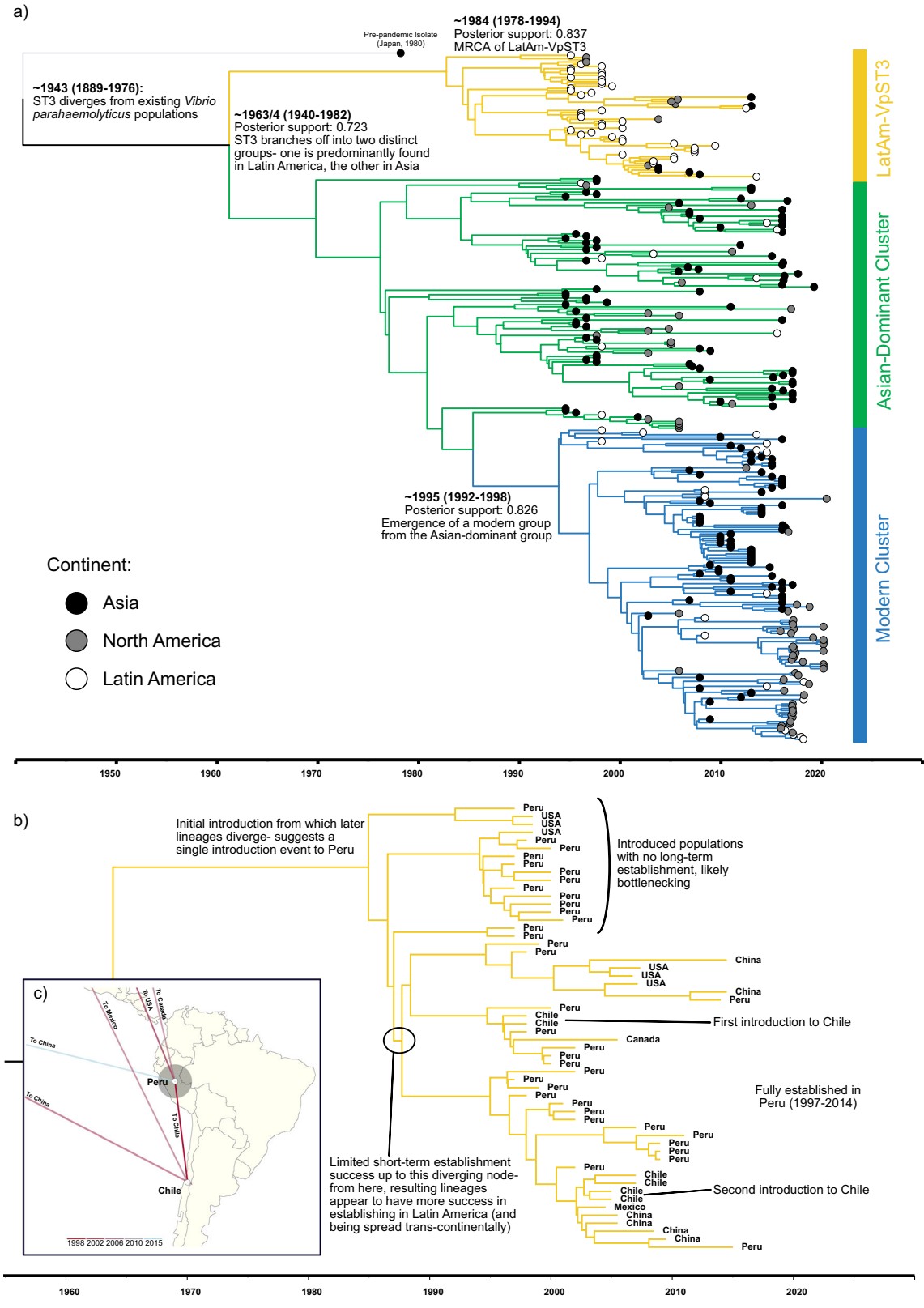

**Fig. 2 | Bayesian phylogenetic reconstruction of VpST3. a** VpST3 Maximum Clade Credibility consensus tree highlighting 3 key clusters within the VpST3 collection, colour coded to isolate origin continent. **b** Specific focus on LatAm-VpST3 phylogenetic reconstruction. **c** Geographic distribution pattern of LatAm-VpST3 from Peru, visualised using SpreaD3[77].

LatAm-VpST3 samples were not found in Asia (the continent from which it is estimated to originate) during the initial global expansion period of VpST3, while it had a dominant early presence in Latin America. Migration models suggest that, in the VpST3 collection, in general, there was greater movement from Asia to South America than the reverse (see Fig. S2). However, the LatAm-VpST3 group itself (Fig. 2b) originated from a single ancestor in Peru, from which most branches bottlenecked with limited long-term establishment success.

One branch resulted in lineages that became established in Latin America from 1997-2014 (posterior support of 0.991) and was introduced regionally to Chile in two separate introductions (posterior support of 0.972), and then trans-continentally to China and the USA (Fig. 2c) with a posterior support of 0.853 and 0.899 on these branches respectively.

Bayesian skyline analysis (Fig. 3a) indicated a sharp increase in effective population size of LatAm-VpST3 from 1985, continuing until around 1995 and levelling off after. This pattern of effective population size curve is characteristic of clonal expansion, with large accelerations of growth, followed by bottlenecks as gene transfer rates slow down. We found a similar significant increase at this time when considering the effective population size of the whole VpST3 collection, suggesting conducive conditions supporting simultaneous population growth. However, the LatAm-VpST3 expansion was preceded by an increase in the wider collection around the mid-to-late-1970s, suggesting the LatAm-VpST3 expansion was separate from an earlier local expansion in Asia, but contributed to the overall population increases observed for VpST3 during this second peak.

Exploring the diversity within populations of early LatAm-VpST3 samples (1996–1999) provided an indication of introduction dynamics, in terms of single or multiple introductions. Hedrick's corrected coefficient of genetic differentiation (GST) for LatAm-VpST3 supported a low genetic differentiation (Hedrick's GST = 0.008, sample size = 18), indicative of an introduction pattern with limited initial diversity. The genetic diversity of this group increased slightly over time (0.013 in 2000–2005, 0.016 in 2006–2010), but remained lower than the other sub-populations (see Table S3), reflective of clonal population structures. For a comparative baseline, the average diversity for each site across the whole ST3 population was 0.038. Early samples (1996–1999) belonging to the Asian-dominant group had a higher genetic diversity than LatAm-VpST3 (Hedrick's GST = 0.024, sample size = 34), with the greatest within-population genetic heterogeneity found for this group, particularly in the late 2000s (Hedrick's GST = 0.055, sample size = 21).

## Genetic Differentiation of LatAm-VpST3

Further analysis confirmed LatAm-VpST3 was significantly different from the Asian-dominant and Modern Clusters. TreeStructure[15] confirmed statistically significant ($p < 0.0001$) support for three distinct clusters which defined the separation used in Fig. 2 (see Fig. S3). We calculated genetic variance between the core genome of LatAm-VpST3 and the rest of the collection using two key statistics: AMOVA, an analysis of molecular variance, and the summary statistic, FST- the fixation index of population differentiation due to genetic structure. These statistics found 6.4% and 5.5% significant ($p = 0.001$) genetic differentiation between the LatAm-VpST3 population and the rest of the VpST3 collection, respectively.

Discriminatory analysis of principle components (DAPC) was used to further identify and describe these genetic clusters. Firstly, an unsupervised approach to infer natural genetic clusters (using a K-means algorithm to define the optimum number of clusters) found 18 clusters within the global VpST3 collection (see Fig. S4), with 1 of these 18 clusters containing all but one of the LatAm-VpST3 isolates (98.2%). This suggests a greater variety within the rest of the collection compared to a very-closely genetically related LatAm-VpST3, but also a clear differentiation separating LatAm-VpST3 from the remaining collection. A secondary supervised approach, providing a parameter of 3 maximum clusters (based on the number pre-defined in TreeStructure) to the DAPC analysis, was consistent with the clustering order in TreeStructure (98.2-98.8% similarity across all 3 clusters, see Table S5), but also highlighted a greater genetic distance separating LatAm-VpST3 from the remaining clusters (Fig. 4a). Similarly, when identifying the presence of admixed individuals with posterior probabilities for multiple waves within the collection, we found there was far more admixture occurring between the other two clusters than with LatAm-VpST3 (see Fig. S5).

We interrogated the DAPC results to reveal key unique single-nucleotide polymorphism (SNP) mutations in the core genome that were identified as significant by the analysis in discriminating LatAm-VpST3 from the rest of the VpST3 collection (Fig. 4b), identifying 3 of interest. The SNP mutations at position 235 and 1010 respectively were predicted to have modifier and low functional effects only (synonymous variants), with little evidence of any tangible impact. However, the SNP mutation at position 957 had a moderate functional impact (missense variant) on the *mnmE* gene- a central tRNA-modifying GTPase.

In terms of evolutionary dynamics driving such differentiation, LatAm-VpST3 had the lowest mutation rate, compared to the other lineages (see Table S4, Fig. S6), of $3.44 \times 10^{-4}$ substitutions per site per year, which corresponds to 0.8 mutations in a year for the whole genome. We predicted the functional effect of each SNP (50,030 total) finding 91.5% of these affected non-coding regions with little impact, while 4.8% had a high or moderate effect on a protein, affecting function or effectiveness. LatAm-VpST3 also had a low recombination rate (r/m ratio of 0.164), with recombination affecting around 0.004% of the genome per year (see Table S4).

## Signatures of selection within LatAm-VpST3

We then explored how selection processes could be driving this genetic differentiation, leading to LatAm-VpST3's distinctive success in Latin America but not in Asia. It is important to concurrently consider the distinct climate differences between the coastal waters of the inferred origin (tropical Asian waters) and the introduced region (western coasts of Peru and Chile), to explore possible selective pressures the bacteria would have had to overcome to facilitate its successful establishment, driving the identified genetic differentiation. Such climate differences can be demonstrated clearly using satellite-derived sea surface temperature (SST), salinity and oceanic pH data collected during the expansion period. A comparison of the marine environments off the coast of Peru and southeast China, the coastlines indicative of the location where the largest number of isolates were discovered in Latin America and Asia respectively, revealed distinct environmental differences (see Figs. S7–S9). Generally, SSTs in coastal areas of southeast China were higher but had a greater range (maximum of 29.22 °C, minimum of 13.97 °C) than Peru (maximum of 26.32 °C, minimum of 15.36 °C). However, Peru had a greater inter-annual variation of SST, characteristic of the El Niño Southern Oscillation (ENSO), where El Niño events (the warm phase) result in more tropical conditions. Peru waters had a higher salinity than those off the coast of China, ranging from 34.9 to 35.4psu compared to China's range of 31.9 to 33.5psu. Additionally, Peru's marine environment had a lower pH generally, ranging from 8.02 to 8.1 (average of 8.06) compared to China's pH range of 8.03 to 8.15 (average of 8.08).

In the core genome, we found evidence for non-neutral selection, and that the evolution of LatAm-VpST3 was mainly driven by purifying selection, indicated by a negative Tajima's D ($-2.62$, $p < 0.01$). This negative Tajima's D is also indicative of population expansions. We then assessed whether mutations in the core genome were associated with particular bioclimatic conditions (referred to here as environmentally-associated SNPs). We used redundancy analysis (RDA) to identify environmentally-associated SNPs within LatAm-VpST3, using SST and salinity data (extracted as binary values representing positive or negative anomalies for the location and date of the sample discovery). Our RDA model found that 2 constrained axes captured 0.5% of the SNP variance, with low predictive capacity expected when most SNP mutations are likely to be neutral. Our significant RDA model ($p = 0.014$) found 121 candidate SNPs that were under selection as a function of the provided environmental predictors. A total of 88 of these SNPs were detected due to positive

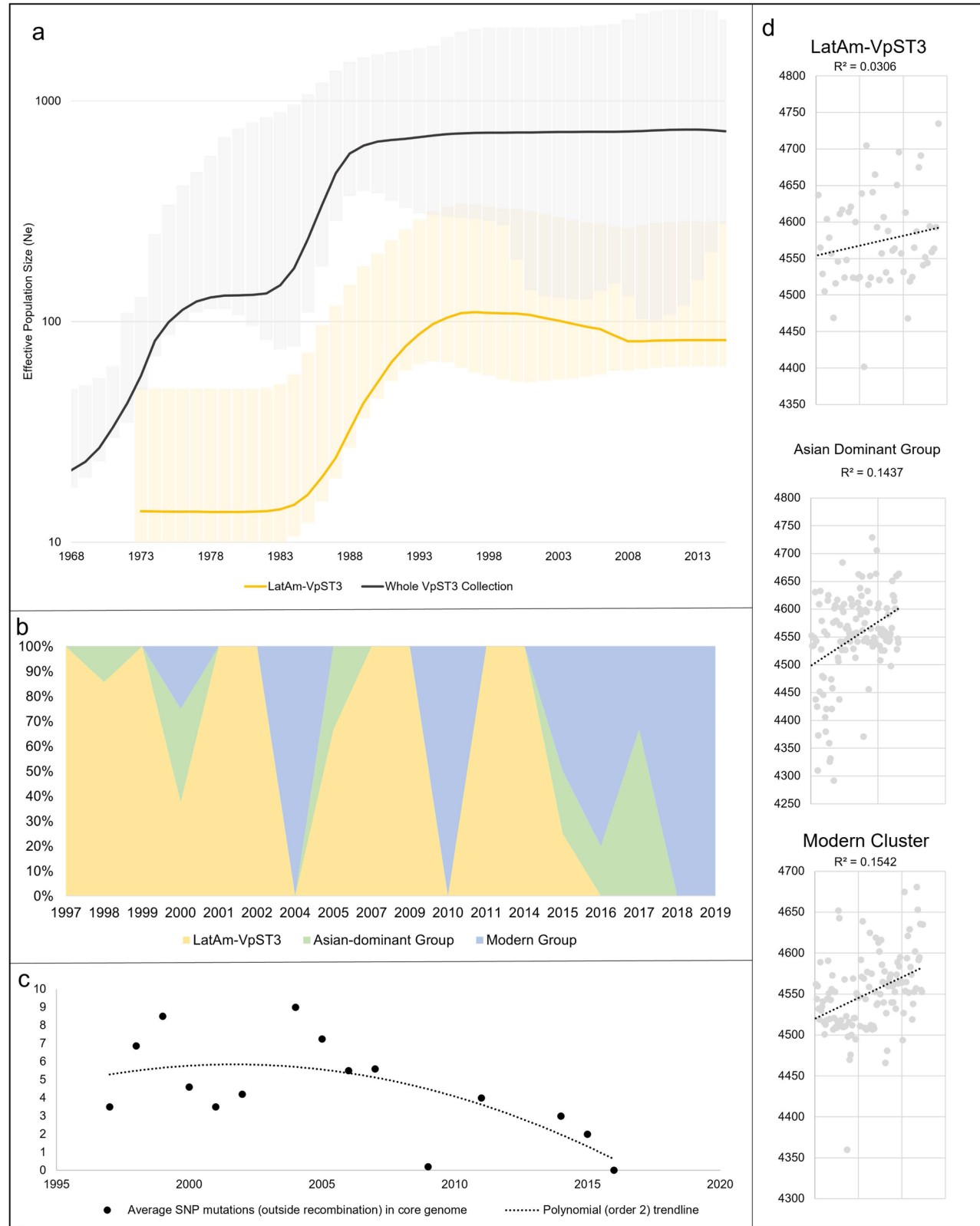

**Fig. 3 | Temporal evolutionary trends in VpST3 collection. a** Bayesian Skyline Plot showing estimated effective population size for LatAm-VpST3, with upper and lower bounds of the 95% highest posterior density interval. **b** Prevalence of each sub-group in Latin America over time shows patterns of shifting dominance. While LatAm-VpST3 remained dominant in Latin America compare to the Asian-dominant group, in recent years LatAm-VpST3 is being replaced with the modern group. **c** Decline in average mutations per core genome in LatAm-VpST3 over time (averaged across isolates found in the same year). **d** Trend in total gene count over time for isolates within each of the three clusters.

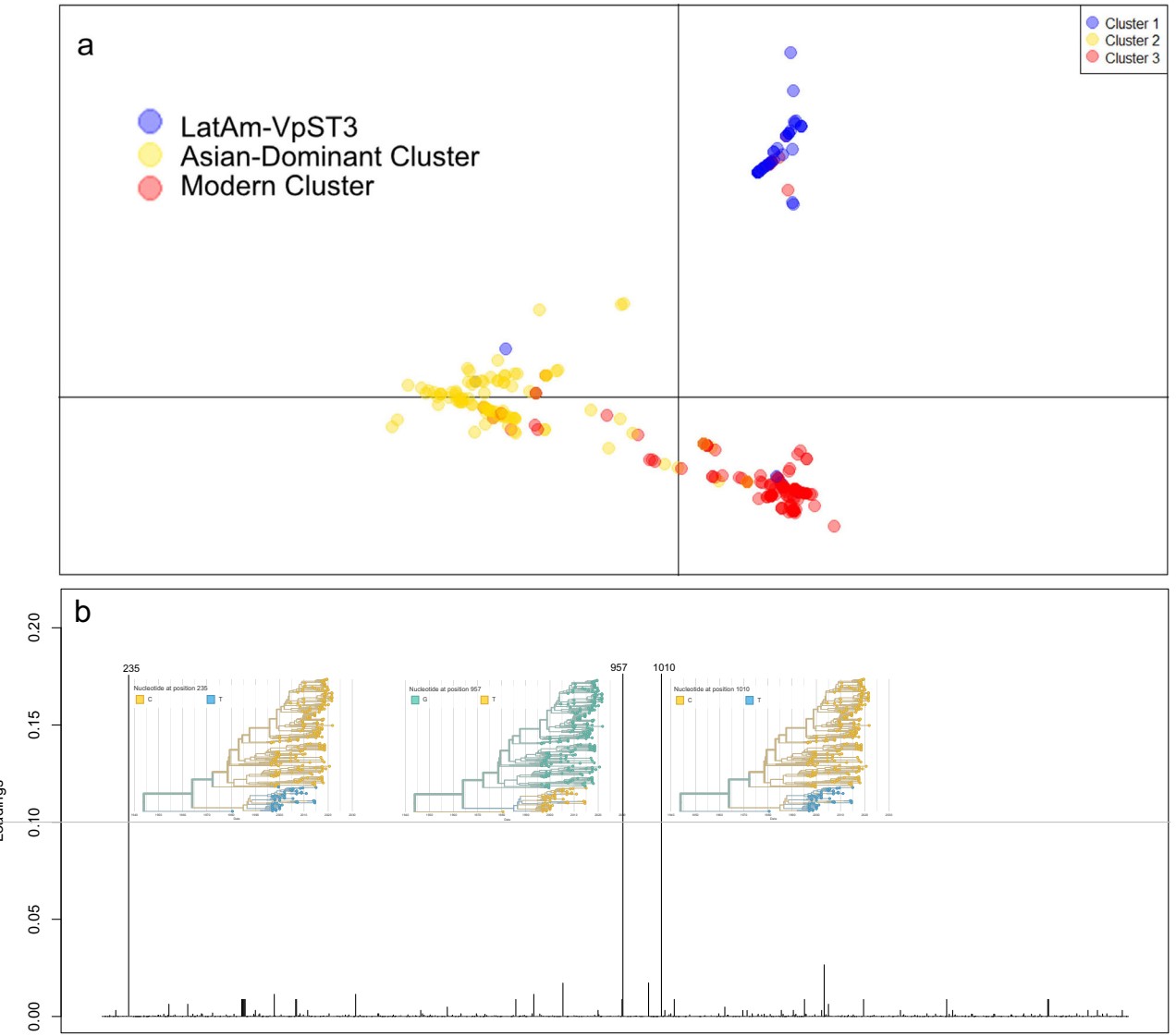

**Fig. 4 | Supervised discriminatory analysis of principle components analysis.**
**a** Scatterplot visually highlighting differentiation and genetic distance of LatAm-VpST3 from the remaining clusters. **b** Loading plot highlighting the position of SNP mutations and the associated allele diversity that were the highest contributors (over a threshold of 0.1) to the distinct evolution of LatAm-VpST3 that discriminates it from the rest of the VpST3 collection.

associations with SST (notably including the non-synonymous SNP at position 957 mentioned previously as a mutation specific to LatAm-VpST3), and 33 with salinity.

We also looked for signatures of selection beyond the core genome by constructing the pangenome, with a particular focus on 'shell' accessory genes- those present in 15–95% of the population. A total of 400 shell genes were found within the whole collection. Using a pangenome-wide association study tool, we identified accessory genes whose presence across the VpST3 collection was significantly correlated ($p < 0.05$) with LatAm-VpST3 (Table 2). A number of these genes had functional roles within putrescine utilisation pathways or provided resilience and adaptation to various environmental conditions such as salinity, nitrogen and pH. LatAm-VpST3 samples have open pangenomes, with a higher average total gene count than the rest of the collection, and a number of 'shell' genes from the entire population were found to be core in LatAm-VpST3 (see Table S6). We then investigated significant associations between the presence of accessory genes and local environmental conditions, in terms of whether certain accessory genes were more (or less) present in years with positive or negative anomalies of both SST and salinity. Notably, we found all the LatAm-VpST3 samples were found in positive salinity anomaly years. Therefore, we could only run this specific environment-association analysis for SST. We identified genes within LatAm-VpST3 that were associated with SST anomalies (see Table S7)- notably 73 genes were present exclusively within positive SST anomaly years (specificity = 1.0), including type VI secretion system baseplate subunit *TssF*, though they were not present in every positive SST anomaly years. These associations were significant ($p < 0.05$), providing support for the presence or absence of these genes being related to SST anomalies. A large number of the genes detected that were related to SST did not have any functional annotation (ie hypothetical proteins).

We used dN/dS ratios (comparing the number of nonsynonymous and synonymous substitutions) to identify sites that were under pervasive selection in identified genes of interest within LatAm-VpST3 (Table S8). Three genes contained sites under significant positive selection ($p < 0.01$); 14 sites of positive selection were identified in a bacterial restriction-modification system specific to LatAm-VpST3, 4 sites within a gene found only in positive SST anomalies, and notably, we found 7 sites under positive selection within *mnmE*, the gene affected by the non-synonymous mutation that was a key

**Table 2 | Accessory gene presence significantly associated with LatAm-VpST3 and functional annotations**

| Gene function, name (where applicable) and chromosome location | Role | Presence inside LatAm-VpST3 | Presence outside LatAm-VpST3 | p-value (Bonferroni-corrected) |
|---|---|---|---|---|
| **Genes that are significantly 100% sensitive in LatAm-VpST3 (always present in this group, not necessarily always present outside)** | | | | |
| Sodium: proton antiporter Chromosome 1 | Involved in regulation of pH and sodium concentration, connected to bacterial adaptation to high salinity[22] | 100% | 41.18% | 3.60E-11 |
| Lactoylglutathione lyase Chromosome 1 | A critical enzyme in methylglyoxal detoxification, previously reported to contribute to survival of Salmonella in nutrient-rich environments[71] | 100% | 43.92% | 3.62E-10 |
| Glutamine synthetase family protein Chromosome 1 | Controls nitrogen homoeostasis by assimilating ammonium into glutamine. | 100% | 44.31% | 9.09E-10 |
| FAD-binding oxidoreductase Chromosome 1 | Catalysers oxidation. | 100% | 44.71% | 9.66E-10 |
| Gamma-glutamyl-gamma-aminobutyrate hydrolase family protein Chromosome 1 | Involved in bacterial putrescine utilisation pathway[72] | 100% | 44.71% | 9.66E-10 |
| Hypothetical protein Chromosome 1 | N/A | 100% | 44.71% | 9.66E-10 |
| *puuR*- HTH-type transcriptional regulator Chromosome 1 | Recombinant protein, involved in putrescine pathways[73] | 100% | 44.71% | 9.66E-10 |
| Hypothetical protein Chromosome 1 | N/A | 100% | 44.71% | 9.66E-10 |
| FAD-dependent oxidoreductase Chromosome 2 | Catalysers oxidation. | 100% | 63.14% | 0.000279 |
| *aguB*- N-carbamoylputrescine amidase Chromosome 1 | Converts N-carbamoylputrescine to putrescine, involved in biofilm production[74] | 100% | 67.45% | 0.003067 |
| *aguA*- agmatine deiminase Chromosome 1 | Involved in putrescine pathway[75] | 100% | 67.84% | 0.0034 |
| **Genes that are significantly specific to LatAm-VpST3 (only seen in this group, absence in the rest of the collection)** | | | | |
| Hypothetical protein Chromosome 1 | N/A | 14.29% | 0% | 0.044542 |
| N-6 DNA methylase Chromosome 1 | Involved in bacterial restriction-modification systems | 14.29% | 0% | 0.044542 |
| Type III restriction-modification system endonuclease | Restriction-modification system- can reduce genetic exchange from other species / clonal complexes and only allow genetic exchange among close relatives[76] | 14.29% | 0% | 0.044542 |

Associations scored using a two-sided Fisher's exact test in Scoary[68], accounting for population structure using a post-hoc permutation test based on a pairwise comparisons algorithm.

discriminator of LatAm-VpST3 from the rest of the collection. Twenty genes had sites with significant negative selection, ranging from 1-10 sites ($p < 0.05$ to $p < 0.01$). These included a pH and sodium regulator gene, and genes involved in putrescine pathways.

**Post-establishment specialism**
After its introduction around 1984, LatAm-VpST3 underwent strong population expansion into the late-90s, peaking in 1997 (Fig. 3a), where isolates were first discovered belonging to this group in Peru. Based on our sample collection, LatAm-VpST3 remained the dominant established VpST3 sub-population in Latin America for around 17 years (Fig. 3b). However, a decline in the effective population size was observed in recent years representing a bottleneck. Previously, we found LatAm-VpST3 to have a low mutation rate ($3.44 \times 10^{-4}$ substitutions per site per year). We found a further temporal trend of declining number of mutations in the core genome per isolate, averaged across those found in the same year, in LatAm-VpST3 over time (Fig. 3c). The average number of core genome mutations increased initially (peaking in 1999), before falling, then peaking one more time (in 2004), before a steady decline into modern years, representing a decline in the pace of genome evolution, when evidence of a clonal replacement starts to emerge. Similarly, we found LatAm-VpST3 to have a low recombination rate (r/m ratio of 0.164), with recombination affecting around 0.004% of the genome per year (see Table S3). When exploring total genes present in each isolate, there was no clear temporal signal compared to the other two clusters which have a weak, but

increasing, trend over time from likely gene uptake (Fig. 3d). The Modern Cluster, which has a higher mutation and recombination rate, is replacing LatAm-VpST3 in Latin America (Fig. 3b). Of the 17 samples collected from Latin America post-2015, only 1 belonged to LatAm-VpST3, but 17 of these (representing 70.6%) were assigned to the Modern Cluster.

## Discussion
Using a range of approaches and genomic analysis of a global collection of VpST3 isolates, and specifically the inclusion of the novel strains from Latin America, has provided insights into the expansion and establishment of VpST3 in Latin America.

**Parallel emergence of VpST3**
Previously, the global expansion of VpST3 was linked to the downstream transcontinental spread from an initial emergence in India in 1996[10]. However, our results suggest VpST3 was already present in Latin America at this time, with the temporal origin of LatAm-VpST3 in Latin America estimated around 1984 (1978-1994). This ancestral variant then underwent a successful adaptation to local conditions over its evolutionary divergence from both this ancestral stage, and VpST3 isolates found in Asia.

Our analysis specifically rules out a possible lineage connection between LatAm-VpST3 and those populations emerging in India in 1996 as they were placed in statistically significantly different clusters within the Bayesian phylogeny. Indeed, the monophyletic nature of

LatAm-VpST3 and the large genetic differentiation would confirm a parallel emergence, rather than an introduction, as they suggest that, either the LatAm-VpST3 population had already existed for long enough to significantly diverge from the Asian-dominant populations, or was genetically distinct from the point of initial emergence. Such a parallel emergence would explain why this group was not detected in Asia pre-2005. Further support for this parallel emergence can be found in the patterns of effective population size in the Bayesian skyline of LatAm-VpST3, with a small bottleneck population seen from the early 1970s before a rapid expansion in the late-1980s/early-1990. While the absence of intermediary ancestral states on the long evolutionary branch to LatAm-VpST3 could infer missing data related to introductions or evolution, the within-population diversity of LatAm-VpST3 was initially low, confirming no evidence for multiple introductions, with VpST3 instead functioning as a background population before the reported emergence in the late 1990s. *Vibrio* are characteristically able to remain in a dormant state in environmental reservoirs until optimum conditions trigger rapid population growth[16].

The possibility that both groups were able to simultaneously emerge to cause outbreaks in 1996-97, on different sides of the globe, is plausible based on the arrival of optimum climate conditions from the global, tele-connected impacts of the exceptional 1997 El Niño event[13]- both in Asia and Latin America. This historically powerful warm phase of the El Niño–Southern Oscillation warmed large areas of the tropical Pacific, increasing suitability for *V. parahaemolyticus* and providing a mechanism for local dispersal (due to the movement of warm waters), likely contributing to the emergence of infections across Peru and north of Chile, concurrently with the epidemic emergence in India, although the variants involved in both cases were different and followed distinctive evolutionary trajectories. The absence of data along the evolutionary branch separating LatAm-VpST3 from the rest of the population skews our representation towards evolutionary outcomes that were successful, omitting earlier prototypes of the evolutionary pathway before the first well-adapted isolate was detected.

The emergence of LatAm-VpST3 is interestingly coincident with the intercontinental introduction of the seventh pandemic El Tor *V. cholerae* to Latin America, dated around 1985-1989, which contributed to an epidemic in 1991[17]. Similarly here, this date precedes reported infections by multiple years. Such temporal lag is expected for *Vibrio* populations; firstly due to the population growth required to result in human transmission to the point of clinical samples being acquired in medical settings, and secondly due to the specific ability of *Vibrio* to lie dormant as viable but nonculturable (VBNC) cells[18] until optimum environmental conditions arrive- in the case of LatAm-VpST3, a strong El Niño event bringing tropical conditions in 1997.

## Signatures of selection to the distinct climate

Our results start to offer suggestions as to why VpST3 was so successful specifically in the distinct climate of Latin America, compared to co-existing Asian populations. Studies have highlighted how genomic evolution may aid pathogenic clone emergence and expansion[4,19], related to non-synonymous core genome substitutions or acquisition of genes. We explore this in the context of Latin America's distinct climate, finding evidence for local adaptation to increase environmental fitness, and evolutionary mechanisms for survival during local dispersal.

Firstly, we found non-synonymous core genome substitutions associated with the distinct climate in Latin America. A key mutation that delineated LatAm-VpST3 from the rest of the VpST3 population was predicted to have a non-synonymous effect on *mnmE*- a gene located on chromosome 1 and essential for growth and pathogenicity. Additionally, this gene has been found to promote survival in more acidic environments[20]. The western coast of Latin America (such as off Peru) is more acidic than those found in tropical Asian waters.

Evidence of positive selection at 7 sites within the gene, and the success of LatAm-VpST3 in a high-acidity region is suggestive this mutation providing some survival advantage in acidic environments. Such an adaptation would not have been necessary for populations emerging in Asia, with this mutation being unique to LatAm-VpST3. This demonstrates a biological mechanism in which a population was positively selecting for an adaptive mutation providing increased environmental fitness in a distinct climate, and differentiating that population to succeed in a new area. Additionally, SNPs were found to have associations with SST and salinity anomalies in a significant RDA model, opening avenues for further investigation of environmentally-associated SNPs. Mavian et al[21]. similarly found mutations potentially involved in adaptation to the aquatic environment, associated with clonal expansion of *Vibrio cholerae* in Haiti, in the form of mutations affecting genes with roles in virulence, survival in different niches and environmental stress responses. Specifically, the SNPs associated with positive SST anomalies highlight the potential role of El Niño events in the evolutionary pathway of LatAm-VpST3- with innovation resulting from the increased SSTs and more tropical conditions present during these events, an environmental variation not experienced to the same extent by populations evolving in Asia.

Secondly, during its parallel emergence, LatAm-VpST3 gene acquisition would have been mediated by a distinct climate from the Asian population. Within the pangenome, we found a significant difference between the presence of a sodium-proton antiporter accessory gene (located on chromosome 1) in LatAm-VpST3, where it was always present, and the rest of the collection where it was only present in 41.2% isolates. This gene is involved in regulating pH and sodium content[22], which similarly suggests LatAm-VpST3 emerged in Latin America because it was well-adapted to its new acidic and saline environment. *V. parahaemolyticus* usually prefers salinities below 34psu, but the functional role of this gene enhancing environmental fitness would facilitate VpST3 survival in the highly saline waters found off the coast of Latin America, particularly during high salinity anomalies. When exploring environmental associations between accessory gene presence and positive or negative climate anomalies, it became apparent that every LatAm-VpST3 isolate was found in years with positive sea salinity anomalies, providing support for a possible high-salinity survival mechanism. Salinity has been identified as a key driving force of *Vibrio*-related epidemiology[3] but this represents a new indication of possible gene-level adaptation to salinity in the environment. Higher spatiotemporal resolution of genomic metadata will enhance our ability to retrospectively reconstruct *Vibrio* population dynamics in the environment and the variables driving these. This includes the future inclusion of further environmental variables associated with evolutionary mechanisms that have more local biochemical dynamics, such as chitin availability from planktonic organisms which can increase environmental DNA uptake[23], or dissolved oxygen which is associated with the presence of virulence factors[24]. While our analysis found particular associations with SST and salinity, such variables could be providing proxy information on other components of the dynamic and interconnected marine ecosystem, from which higher-resolution analysis is required to isolate specific environmental associations. In the future, live strains should be collected to facilitate gene function experiments which are necessary to confirm the associations identified here in real-time.

Alongside these environmental tolerances, we found evidence of features that may have facilitated local dispersal of VpST3 in Latin America, related to putrescine pathways and biofilm formation. Four genes located in chromosome 1 and involved in bacterial putrescine pathways (*aguA*, *aguB*, *puuR* and an unnamed gamma-glutamyl-gamma-aminobutyrate hydrolase family protein) were found to be significantly more present in LatAm-VpST3 (100% presence) than the rest of the collection (44.7–67.8% presence). Putrescine has a role in modulating biofilm formation in environmental Gram-negative

bacterium such as *Shewanella oneidensis*[25], which offers biofilm-mobilisation mechanisms, in which bacteria are able to 'hitchhike' on a range of organic and inorganic matter in ocean currents[26], similarly found within environmental isolates during the successful clonal expansion of *V. cholerae* into Haiti[21]. Additionally, living within a biofilm can actually offer a range of further benefits to *Vibrio* bacteria, including increased metabolic responses and functional diversity[27]. These genes were not specific to LatAm-VpST3, as they were present in other groups that would have similarly required dispersal mechanisms, however their ubiquitousness in LatAm-VpST3 samples highlights a likely contribution to LatAm-VpST3's emergence. Though evidence of purifying selection was found within these genes, we would expect them to be under purifying pressure after fulfilling their function of facilitating introduction. Our overall Tajima's D of −2.6 is consistent with that found for clinical *V. cholerae* samples from the clonal expansion into Haiti (−2.3[21],). This post-emergence purifying signal is likely following intensive adaptation of early emergers[28]. Studies have identified environmental isolates, as opposed to clinical isolates, to be more likely driven by diversifying selection[21] however the majority of the LatAm-VpST3 samples were from a clinical origin making it difficult to ascertain whether a similar pattern would be exhibited in Latin America.

While exploring the specific evolutionary role of each chromosome was outside the scope of this study, most of the accessory genes that emerged as specific to the LatAm-VpST3 population, or associated with environmental variables, were located on the first chromosome, with the exception of a FAD-dependent oxidoreductase gene. This chromosome is the largest of the two, and impacts of mutation and recombination have previously been found to be higher in this chromosome for *V. parahaemolyticus* ST36[4], however recent genomic insights suggest the second, smaller chromosome may be involved in adaptation to different ecological niches[29,30]. While this was not identified here, specific analysis for each of the two chromosomes (with specific parameters set for each chromosome) would provide further insights into its role in facilitating the expansion of VpST3 in distinct marine environments.

## Establishment and Isolation

Our results provide evidence for the successful establishment of LatAm-VpST3, having been recorded in Latin America for almost two decades, which is unique for a strain of a dynamic bacterium like *V. parahaemolyticus*. During this time, LatAm-VpST3 became specialised with limited innovation, evidencing a well-adapted population that has reached the peak of environmental fitness in the adaptive landscape, compared to heterogeneous recently-introduced populations undergoing diversification and requiring innovation. Our analyses found LatAm-VpST3 had no trend of gene uptake compared to the rest of the collection and a far lower rate of recombination (r/m = 0.164, affecting 0.004% of the genome). This is lower than recent estimates of recombination in *V. parahaemolyticus* populations, with studies estimating recombination affects 0.017% of *Vibrio* genomes per year[31], however is similar to the clonally evolving *V. cholerae* 7th pandemic lineage (r/m = 0.1[32];), and characteristic of clonal evolution[33]. Restriction-modification system genes, which limit exchange of genetic material, were found in 14.3% of the LatAm-VpST3 samples, and nowhere else. This cements the isolation of LatAm-VpST3 from other *Vibrio* populations, exacerbated by genetic differentiation, due to no genetic interaction with other populations. Additionally, we observed a decline in the number of mutations in the core genome. While the whole VpST3 collection fell in the lower end of mutation ranges previously reported for other *Vibrio* clonal groups[4,34,35], LatAm-VpST3 had a particularly low mutation rate. Temporally, core genome mutations increased during the initial expansion of LatAm-VpST3, likely as conditions returned to normal after the 1997 El Niño event, resulting in negative SST anomalies and positive salinity anomalies,

requiring adaptive environmentally-driven mutations. Subsequent environmental conditions were more uniform, reducing the need for further adaptation or innovation, explaining the decline in mutations thereafter, leading to a bottleneck.

This high level of specialisation in the Latin American marine environment may explain the lack of success of LatAm-VpST3 samples outside of Latin America. Additionally, after decades without competition or need for innovation, this may put LatAm-VpST3 at a disadvantage against the modern cluster, leading to its decline. The modern cluster replacing LatAm-VpST3 is generally successful in all continents, across vastly distinct environmental conditions, suggesting it is more of a generalist. With much higher mutation and recombination rates, it is likely capable of adapting to a range of changing environments, which is distinct from the current evolutionary mechanisms of LatAm-VpST3. We propose that LatAm-VpST3 was favoured within the particular ecological niche in Latin America but has likely become ecologically isolated due to a lack of gene exchange and diversification. Such isolation has been observed within *Vibrio* populations relating to host niches[19], but the consequence here is likely to result in clonal replacement.

In conclusion, the emergence of VpST3 in Latin America was therefore not a transcontinental spread of the epidemic strain emerged in Calcutta in 1996, but was instead the result of an ancestral VpST3 variant introduced into Latin America before this, where it underwent a successful adaptation to the distinct, local environmental conditions over its evolutionary divergence from this ancestral stage, differentiating these populations from VpST3 isolates found in Asia. The arrival of the exceptional warming conditions in 1996-97 globally associated with the strong El Niño event drove the emergence of infections across Peru and north of Chile, concurrently with the epidemic emergence in India, with distinct variants involved in both regions, following differentiated evolutionary trajectories. Ultimately, the addition of novel Latin American samples helped to uncover this new angle to the global expansion of VpST3, highlighting the importance of surveillance data for a classically underreported bacterium. To fully understand environmentally-driven selection, future genome submissions should be accompanied with high-resolution spatio-temporal data[36], alongside in-situ experiments testing adaptive responses to environmental stimuli, on a much finer resolution than coarse satellite data, which will facilitate characterisation of pathogen eco-evolutionary responses to environmental changes. We have taken the unique opportunity to combine environmental and evolutionary approaches to explore the complex dynamics surrounding the pandemic expansion of a bacteria strain into new and distinct climates, to gain a greater understanding into clonal expansion and global spread of pathogens. Continuing to endeavour to understand the dynamics of pandemic expansion is critical to pre-empting and mitigating future pandemic health threats.

## Methods

### Sequencing of novel isolates

A total of 32 novel VpST3 strains from the National Centre for Public Health in Peru, collected between 1997-2007 from regions spanning Iquitos in Northern Peru, to Puerto Montt in southern Chile, were sequenced using MiSeq Illumina (denoted in Supplementary Table 1). Strains were submitted from regional laboratories to the reference center at the Instituto Nacional de Salud (Lima, Peru), with only information about the location and date of isolation provided. This study was conducted within the framework of the National Surveillance for Acute Enteric Diarrhoea approved by the Instituto Nacional de Salud of Peru and the Committee of Research and Ethics approval was waived in accordance with the national legislation and the institutional requirements for Public Health Surveillance (Ministry Resolution N ° 730-2022-MINSA). The strains were retrieved from a − 80 °C storage freezer, transferred to Luria-Bertani (LB) medium (with 3%

NaCl), and incubated at 37 °C with shaking at 250 rpm. The genomic DNA was extracted with the DNeasy Blood and Tissue kit (Qiagen, Valencia, CA) from overnight cultures. The DNA concentration was determined using a Qubit double-stranded DNA BR assay kit and a Qubit fluorometer (ThermoFisher Scientific, Waltham, MA), according to manufacturer's instructions. Sequencing libraries were prepared with the Nextera XT DNA sample preparation kit (Illumina) and subsequently sequenced using an Illumina MiSeq. The assembled sequences were submitted under BioProject PRJNA1062747. All samples had a quality (QC) above Q30 (Table 1).

## Genomic data and processing

The new isolates were added to an initial global collection of 434 publicly-available VpST3 genomes (see Supplementary Table 1) for genomic analyses. Strains were selected from global databases based on two criteria; the samples were accompanied with sufficient metadata (with a sample collection date at a minimum annual resolution, and a location attribute with a minimum country resolution). The sequence type of each genome was obtained using multi-locus sequence typing (MLST) based on seven housekeeping gene loci across both chromosomes (*dnaE, gyrB, recA, dtdS, pntA, pyrC* and *tnaA*) from the *V. parahaemolyticus* scheme in the PubMLST database[37] using default settings in MLST v2.11 (available at https://github.com/tseemann/mlst)[37]. This included 16 genomes recently made publicly available[38,39], and not yet utilised. We downloaded the raw sequence data (in fastQ format) for each VpST3 sequence, which were then run through the same analysis pipeline to ensure consistency. The raw sequence data was processed, validated, quality-filtered, and annotated using the Bactopia v2.0.2 pipeline with default parameters[40]. Reads that failed to pass length or quality requirements in the quality check module of the Bactopia v2.0.2 pipeline were filtered out and excluded from downstream analyses. This pipeline then assembled the sequences with default parameters in Shovill v1.1.0 (https://github.com/tseemann/shovill), a standard de novo genome assembler for Illumina paired-end reads, with assembly quality statistics generated using QUAST v4.6.0[41]. The trimmed VpST3 collection contained 312 isolates in total from 19 countries within 3 continents (163 from Asia, 71 from South America and 78 from North America), including a 'pre-pandemic' strain[42], isolated in Japan in 1980, to explore relatedness to the pandemic clone ST3. Parsnp v1.5.6[43] was used to identify core SNPs (single nucleotide polymorphisms) across all sequences, and create a core genome alignment using the reference genome *Vibrio parahaemolyticus* RIMD 2210633. Recombining regions were identified and removed using Gubbins v3.1.6[44], alongside any sequences that were too similar, to provide the final nonrecombining core genome alignment. SNP calling was performed using snp-sites v2.5.1[45], based on the same reference genome, which were subsequently annotated using SnpEff v5.1[46] to predict functional effects on genes. We constructed a maximum-likelihood phylogenetic tree from the non-recombining SNPs in IQ-TREE v2.2.3[47], with 1000 normal bootstrap replicates and a TVMe+ASC substitution model (a transversion model with equal base frequencies and ascertainment bias correction, identified as the best-fit model according to a Bayesian Information Criterion analysis), rooted to the pre-pandemic strain, in which 818 sites of 2334 were found to be parsimony-informative.

## Bayesian phylogenetic analysis

Temporal signal and conformation to a molecular clock was confirmed in TempEst v1.5.3[48]. VpST3 dynamics were reconstructed using a Bayesian phylogeographic approach in BEAST v2[49], a package that facilitates the statistical inference of historic population dynamics and evolutionary parameters from molecular sequence data using a Markov chain Monte Carlo (MCMC) algorithm sampling from a posterior distribution. A structured coalescent was applied to the non-recombining core SNP alignment, using tip dates and discrete traits (as

part of the MultiTypeTree template) to append temporal and spatial information (continent-level) respectively. A relaxed log normal molecular clock model was selected based on previous path sampling and stepping-stone sampling analysis that identified this molecular clock model as the best fit for *V. parahaemolyticus*[4]. The model used a GTR (general time reversible) substitution model with a substitution rate prior (ucld.mean) set to a normal distribution, as used previously for *V. parahaemolyticus*[31] and identified here as having the strongest Bayesian posterior probability support using the BModelTest model comparison implementation[50]. Population sizes and migration rates were both estimated within the model, with an exponential prior set, and sensitivity analyses for convergence was conducted to set remaining parameters and priors. The model was run for 250,000,000 states in total. Tracer v1.7.1[51] was used to confirm the convergence of the MCMC outputs (ESS > 200) and provide summary statistics. TreeAnnotator, a BEAST2 tool, was used to generate a final Maximum Clade Credibility (MCC) tree, annotated with estimated divergence dates and likely geographic origin of the most recent common ancestor. We measured the phylogenetic signal (K statistic) of geographic traits (both country of origin and continent), using picante v1.8.2[52], to test for a geographic signal of phylogenetic clustering within our tree. This stage of the analysis led to the identification of the LatAm-VpST3 sub-group, providing the focus of subsequent analyses.

## Population genomics

We tested for structure within the population using the non-parametric, statistical approach in TreeStructure v0.1.0[15], performing 100,000 tree simulations, set to a significance threshold of $p < 0.001$. Discriminant Analysis of Principle Component (DAPC) was applied to identify genetic clusters and differentiation using the adegenet package v2.1.10[53], in R[54]. Firstly, natural clusters were identified using no prior information, with a k-means clustering algorithm identifying the optimum number of clusters as 18, giving the lowest Bayesian Information Criterion value. We then ran DAPC using the 3 pre-defined populations found in TreeStructure; with 100,000 simulations set at a significance threshold of p < 0.001. The results were interrogated for evidence of admixture, and identification of key SNPs used to discriminate the clusters. Further statistics of genetic differentiation of LatAm-VpST3 from the rest of the collection included calculating the Analysis of Molecular Variance (AMOVA) using the poppr package v2.6.1[55], and calculating the pairwise fixation index (FST) between populations using the dartR package v2.9.7[56], both in R. Effective population size dynamics of LatAm-VpST3 were estimated using a Coalescent Bayesian Skyline in BEAST2[57], utilising a MCMC sampling procedure (ran for 50,000,000 states to reach convergence) to group coalescent events into time intervals and reveal demographic signatures over time. Hedrick's GST (coefficient of genetic differentiation), a standardised version of Nei's Genetic Diversity Index[58], was used as a descriptive statistic to represent within-population genetic diversity. It was implemented using vcfR v1.15[59] in R. Hedrick's GST infers the distribution of allele frequencies for each site, in each of the subpopulations identified, which was averaged across to give an indication of diversity across the whole core genome within each subpopulation, and compared across temporal 5-year bins.

## Oceanic climate data

Historic oceanic climate data was acquired for temporal periods aligning the estimated introduction and divergence of VpST3. SST data were acquired from ERA5 reanalysis which uses the HadISST2 dataset pre-2007[60] for 1982-2010. Salinity data were acquired from Met Office Hadley Centre EN4.2.2 quality-controlled ocean data[61,62] for 1995-2010. Seawater pH data were acquired from the OceanSODA global gridded dataset[63] for 1982–2010. The data were processed using xarray[64] in Python to clip to areas of interest (Peru coast and southeast China

coast to represent the LatAm-VpST3 group and co-existing dominant Asian population respectively), and extracted to form monthly time series and seasonal signals. Data were also aggregated to seasonal averages for each year and appended to each sample based on the isolate collection date and origin country, for climate association analyses.

## Identifying signatures of adaptation

Redundancy analysis (RDA) was used to test for multilocus adaptation in the core genome using the R packages adegenet v2.1.10[53] and vegan v2.6.4[65]. This multivariate technique allowed us to analyse genotype-environment associations to detect loci under selection[66]. Candidate SNPs and their loading values were identified on each RDA axis and then the correlation between these and various environmental predictors (average, maximum and minimum SST and salinity for each season) were calculated.

We also explored signatures of adaptation outside the core genome. Roary v3.13.0[67] was used to construct the pangenome and annotate the presence of accessory genes present within the whole collection. Particularly, we were interested in shell accessory genes (present in 15-95% of the population) that might be indicative of selective adaption. Scoary v1.6.16[68] was used to identify accessory genes whose presence was statistically associated with the LatAm-VpST3 group specifically, compared to the rest of the collection. Scoary was then used again to identify any accessory genes within the LatAm-VpST3 pangenome that were associated with climate anomalies.

To test for selection signatures, we calculated Tajima's D statistic[69] for LatAm-VpST3, which provides a measure of deviation from neutral evolution, indicating if there is significant evidence of purifying or diversifying selection. To reveal site-specific selection pressures in genes of interest (those where previously environmental associations had been established), we estimated dN/dS ratios using Single-Likelihood Ancestor Counting (SLAC) within HyPhy v2.5.48[70], to detect sites under diversifying or purifying selection. SLAC uses a combination of maximum-likelihood (ML) and an adjusted Suzuki-Gojobori counting method to estimate the nonsynonymous (dN) and synonymous (dS) substitution rates on a per-site basis. Significance was calculated for each site using an extended binomial distribution-we set this at $p < 0.05$ or below to consider a site under pervasive selection.

## Reporting summary

Further information on research design is available in the Nature Portfolio Reporting Summary linked to this article.

## Data availability

Sequence data generated during the study is available under BioProject number PRJNA1062747 [https://www.ncbi.nlm.nih.gov/bioproject/1062747]. Previously published sequence data used in this study can be found at the following BioProjects at https://ncbi.nlm.nih.gov/: PRJEB39490, PRJNA176634, PRJNA176652, PRJNA188204, PRJNA215961, PRJNA222557, PRJNA231221, PRJNA233509, PRJNA245882, PRJNA266293, PRJNA273159, PRJNA275536, PRJNA304021, PRJNA306401, PRJNA324079, PRJNA324080, PRJNA324095, PRJNA324096, PRJNA324105, PRJNA325137, PRJNA325139, PRJNA33625, PRJNA345099, PRJNA347505, PRJNA350230, PRJNA357947, PRJNA357986, PRJNA360, PRJNA393608, PRJNA437553, PRJNA437554, PRJNA437555, PRJNA437557, PRJNA437561, PRJNA437563, PRJNA437564, PRJNA438219, PRJNA483379, PRJNA487159, PRJNA531481, PRJNA556706, PRJNA602337, PRJNA613630, PRJNA633360, PRJNA639932, PRJNA643807, PRJNA664306, PRJNA677930, PRJNA706389, PRJNA747744, PRJNA754786, PRJNA754787, PRJNA754788, PRJNA754789. Specific accession codes and metadata can be found in Supplementary Table 1.

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

## Acknowledgements

This work was supported by: Natural Environmental Research Council [Grant number NE/S007210/1] (A.M.C.). Centre for Environment, Fisheries and Aquaculture Science (Cefas) internal Seedcorn funding (A.M.C.). Ministerio de Ciencia e Innovación (Spain) Grants PID2021-127107NB-I00 (J.M.U.). Generalitat de Catalunya Grant 2021 SGR 00526 (J.M.U.). European Union's Horizon Europe research and innovation program under Grant Agreement No.101057554 (IDAlert) (J.M.U.).

## Author contributions

A.M.C., J.M.U., C.H. and R.vA. designed research; A.M.C. performed research; J.M.U. and R.G.G. generated data; A.M.C. analysed data; A.M.C., J.M.U., C.H., R.vA., M.A.M. and C.Y. interpreted results; and AMC and JMU wrote the paper.

## Competing interests

The authors declare no competing interests.
