## [Peer Review File · Nature Communications]

Evolutionary dynamics of the successful expansion of pandemic *Vibrio parahaemolyticus* ST3 in Latin AmericaREVIEWER COMMENTS

Reviewer #1 (Remarks to the Author):

The authors present an extensive genomic eco-spatio-temporal phylogenetic investigation on *Vibrio parahaemolyticus* to unravel the link between the ecosystem (water metadata), country, and their date of isolation. Next to the supplementation with 32 new WGS datasets, databases were mined for VpST3 strains to supplement current analyses. While the authors have presented the work very neatly and provide great visual cues, there are some major questions that arose and should be solved prior to publication:

1. Some sentences require revisions due to their length and/or them being difficult to follow for the reader. Some examples are given below;
2. Some important information is lacking on the DNA isolation along with the QC of the 32 new *V. parahaemolyticus* genomes as presented in the manuscript;
3. Some questions arose on the used tools at different steps in the analyses, along with the potential impact on the downstream analyses, results, and conclusions (see details below);
4. The authors have done various analyses that rely on proper model and coalescent selection. Though, it was not clear if any model/coalescent selection approaches (e.g., using MLE) were applied to reassure the right models were applied;
5. Finally, I believe the manuscript would highly benefit and give even further insights when the authors would consider the (important) factor that the *V. parahaemolyticus* genome is comprised of two individual chromosomes. Inclusion of this pivotal piece of information throughout the manuscript would shed potential important extra insights and elevate the manuscript even further. More detailed reviews and questions are given below:

L26 - We found a VpST3 population (LatAm-VpST3) was introduced > We found a VpST3 population (LatAm-VpST3) that was introduced

L59-66; L75-77 – These sentences require some rewriting to read more fluently and make its content clearer to the readers.

L109-110 – Please include the number of new isolates added here as well as the number of publicly available genomes that were used in current analyses.

L498-511 – Please include the used barcoding approach for the MiSeq sequencing. How was the DNA obtained, what was the QC of the DNA?

L514-535 – Some extended questions on this paragraph:

- Which settings were used in the MLST software and how did the authors handle samples that have submission dates instead of actual sampling dates in the field of selection for their metadata.
- Where the original manuscripts consulted for additional verification which is important for any eco-spatio-temporal genomic analyses?
- Which tools (i.e., software, versions, settings) were used to assemble the genomes?
- Why was a QC filtering of Q10 applied for Illumina MiSeq reads while the median raw read accuracy aims at Q30 reads, which is also intrinsic for QC reporting within the MiSeq device?
- Also, reads of 1kb seems not to be a filtering at the level of raw reads but rather at the level of contigs here?
- Which QC was performed on the the final genome assemblies (i.e., genome completeness/contamination)? Why was no assembly QC implemented in the choice of genomes from the databases?
- The authors write that annotation was performed within Bactopia, though also with PGAP (L503)?
- How does Gubbins handle highly contiguous sequencing data and how does this impact any further data analyses?
- How do these tools handle bacterial genomes that have multiple chromosomes, as *Vibrio* sp. has two chromosomes that might have different evolutionary pressure? It seems this is not considered in current manuscript, which seems an important aspect when working with *Vibrio* sp.
- What is the potential impact of using different tools to get to SNPs and why can the same software not be used for both (cf. Parsnp Versus snp-sites and SnpEff). How does this relate to outputs from software tools such as Snippy (<https://github.com/tseemann/snippy>) and roary (<https://sanger-pathogens.github.io/Roary/>), which present more holistic approaches.
- Was normal bootstrapping or ultrafast bootstrapping applied for the IQtree command? How many valid sites remained after processing and how was the redundancy? Was any down sampling required or performed?

L539-555 – How did the authors confirm the choice of their molecular clock model and coalescent?

Was any model selection approach, using MLE performed to reassure the right models were applied (https://beast.community/model_selection_1 and https://beast.community/model_selection_2). Without proper model selection the authors cannot guarantee the best fit model was applied for current analyses. What was the burn-in percentage that was used? Were any independent runs run and combined. The latter is often recommended as it increases the power of probabilities as individual/independent runs are run with different random number seed that increases robustness when run over different runs and combined afterwards using LogCombiner for example.

L559-579 – What was the burn-in percentage that was used? Were any independent runs run and combined (see comment above).

L156-163 – In this section it is important to get more insights in the actual probabilities to guide the readers in the confidence of the drawn trees and maps. What is the support for the spread to China and other countries as indicated in Figure 2C?

L165-174 – This might also be biased by the used coalescent, which is the Bayesian skyline here. How did the authors test for the proper choice of coalescent to fit the population here? What about the Bayesian Skygrid coalescent model with Hamiltonian Monte Carlo sampling? This allows a more “flexible” fit of the available data to represent the population dynamics. This approach also allows to specify cut-off date and population sizes within the MCMC sampling procedure.

L222-228 – What is the confidence on these recombination rates (cf. comment in M&M section)? What is the impact of gene deletions/inserts besides SNPs? Applying different software (e.g., snippy) would also report these indels for functional assessment as this is the “focus” of this section. It would be of high value to differentiate genes based on their coding chromosome to see if one or the other chromosome has a different evolutionary pressure or not. I believe implementation of this, would shed interesting depth and (new) insights to the understanding in current manuscript. Even performing some BEAST analyses on the separated chromosomes would improve the manuscript.

Reviewer #2 (Remarks to the Author):

This research using a global collection of clinical and environmental VpST3 isolates, based on Vp genome evolutionary analysis, climate associated gene function analysis and oceanic climate changing data found out Vp evolutionary mechanisms driving global expansions of pathogen strains. The finds reshaped the current understanding of the global expansion of the *V. parahaemolyticus* clone and help us understand function of factors like climate, gene mutation in the pathogen expansion. However, we have a couple of comments to address:

1. Only 32 VP strains from Latin, weather these strains well represent the Latin VP? and we want the author describe all the VP outbreaks cases in Latin countries, and demonstrate these 32 strains can represent the Latin VP ST3. All the 32 strains were collected from 1996-2005, it may be hard to collect the strains from decades ago, but why not enroll the strains after 2005, no outbreak cases after 2005?

2. Line160-163 “LatAm-VpST3 itself (Figure 2b) originates from a single ancestor in Peru, from which most branches bottleneck with limited long-term establishment success, while another results in lineages that became established in Latin America (from 1997-2014) as well as being introduced regionally to Chile (in two separate introductions), and then trans-continently to China and the USA (Figure 2c)”. The author tried to clarify the ST3 dissemination routes among Latin-China-USA. But we think the dissemination routes may not be a single way, because the infection sources and the genetic differentiation strains are not occurred in an only unique place. Like the new detected serotype O10:K4(ST3) we firstly reported in Guangxi, south China, but quickly, the north of China detected the O10:K4 as well. It is not likely transmit from south to north, it more likely that the O10:K4 exist in China but the identified time is different. Besides, in this study, lacking information of the VP single ancestor strains from Peru, and from the Fig2 the Latin American, North American and China ST3 strains distribute in the three clusters, indicates the infectious sources complicates, so the author got the conclusion that the VP single ancestor in Peru then transmit to Chile, China and USA need more supports.

3. The ocean temperature and alkalinity (pH 8.06 and 8.0) variation between Peru and China are small. Do these small differences in environment cause genetic mutations or require genetic mutations to adapt?

4. The authors using a pangenome-wide association study tool to find the sea surface temperature (SST) relative genes, and tried to elucidate the Latin-ST3 adaptation mechanism. It found out some adaptation genes may associate with the SST, but lacking the experiment that proved these genes have the function in adaptation. Add a gene function experiment can well validate these genes function and confirm the speculation.

5. In this study, lack of VP ancestral strains from Latin America and all the VP strains used in this study were collected after 1996. So in the evolution analysis, only based on the SNP analysis, it is hard to get the conclusion that the ST3 pathogen introduction in Latin America in 1985.

6. Suggesting listing more detailed genetic information of the 32 newly ST3 strains of *Vibrio parahaemolyticus* from Latin. And show more differences between strains prevalent in Asia in 1996.

7. In this study, some bioinformatics tools were used to calculate and analyse the evolutionary relationship. Like the strains in Fig2 1a are all collected after 1997, how it can draw the result that event occurred before 1940-1998. Most of the results in this paper based on the evolutionary analysis. The author should fully discuss these tools and models used can derive the correct speculation or results.

Reviewer #3 (Remarks to the Author):

Overall, a well-conceived and executed study.

While there may be an association of SNPs with SST, that is not surprising given that is the main variable examined. Caution making too strong of conclusions when other environmental factors (dissolved oxygen, etc.) that may influence genetic change were not examined. This possibility should be acknowledged and appropriately discussed.

Reviewer #1

The authors present an extensive genomic eco-spatio-temporal phylogenetic investigation on *Vibrio parahaemolyticus* to unravel the link between the ecosystem (water metadata), country, and their date of isolation. Next to the supplementation with 32 new WGS datasets, databases were mined for VpST3 strains to supplement current analyses. While the authors have presented the work very neatly and provide great visual cues, there are some major questions that arose and should be solved prior to publication.

We thank the reviewer for providing constructive comments on our manuscript. We have provided responses to clarify the questions that arose including providing further detail on quality control of our novel samples, further methodological detail particularly around model selection and set-up of our Bayesian phylogenetic analysis and added further insights into the chromosome-specific throughout our results and a discussion of the implications of this for further research.

1. Some sentences require revisions due to their length and/or them being difficult to follow for the reader. Some examples are given below;

We have revised particular lengthy sentences to increase ease of understanding for the reader, including in the specific lines identified- 59-66, 75-77 (detailed below under specific questions a and b).

2. Some important information is lacking on the DNA isolation along with the QC of the 32 new *V. parahaemolyticus* genomes as presented in the manuscript;

We have added a table in the main text summarising the extra information and QC of the new genomes presented in the manuscript (Table 2). Specific responses to the further points raised on this can be found below (d, h and i).

3. Some questions arose on the used tools at different steps in the analyses, along with the potential impact on the downstream analyses, results, and conclusions (see details below);

We have clarified particular points around the analysis by providing further information on parameters and settings used within software, model selection processes and how these relate to our results. Specific answers can be found in questions e, g, j, k, l, m, n, and o.

4. The authors have done various analyses that rely on proper model and coalescent selection. Though, it was not clear if any model/coalescent selection approaches (e.g., using MLE) were applied to reassure the right models were applied;

Thank you for bringing this to our attention and giving us the opportunity to add further information on the choices behind our model selection and set-up. We have provided full details in the specific questions below (p, s) on the choices of molecular clock, coalescent model and substitution model to ensure these were the best fit, and how we have added this information into the manuscript.

5. Finally, I believe the manuscript would highly benefit and give even further insights when the authors would consider the (important) factor that the *V. parahaemolyticus* genome is comprised of two individual chromosomes. Inclusion of this pivotal piece of information throughout the manuscript would shed potential important extra insights and elevate the manuscript even further.

We agree this is a fantastic idea and would be the next step in understanding the complex dynamics of *V. parahaemolyticus*, which is becoming a focus of our future research. Our previous work on *V. parahaemolyticus* (published in mBio, Martinez-Urtaza et al., 2017) found the larger Chromosome 1 to be more influenced by recombination. In work before this paper, we also found there was a greater influence of recombination in Chromosome 1 (See Figure 1, with the high frequency of orange SNPs in recombining regions in Chromosome 1). We agree if we were focusing on recombination in this study, chromosome-specific analysis would be necessary to explore these distinct evolutionary effects. However, in this study the effects of recombination were removed to facilitate molecular clock phylogenetic analysis, and therefore we were focusing on SNPs in non-recombining regions. Previous studies have found phylogenies to be consistent across both *Vibrio* chromosomes, allowing phylogenetic conclusions to be applied equally to both chromosomes (Kirkup et al., 2010). Our early work (Figure 1) found the frequency of SNPs in non-recombining regions was not largely different between the two chromosomes, which was also found in our previous work on *V. parahaemolyticus* (Martinez-Urtaza et al., 2017) which similarly found no large difference in the frequency and distribution of SNPs per length between the larger first chromosome and smaller second chromosome. Finally, this study is focused on variations at a population level rather than a genomic level, with all the analysis based on the use of a core alignment for the whole genome, without splitting the data into two chromosomes. This is the typical approach for these analyses, with the division of sequencing data in different chromosomes rarely considered, which is why we agree this unexplored aspect should be considered in future studies.

To acknowledge the importance of the two individual chromosomes, we have added in-text mentions of whether the genes of interest are located in Chromosome 1 or 2 (described fully in answer to question u below) but regret we cannot take this analysis further at this stage without the study becoming too broad and unfocused. We have also added a discussion of how the presence of two chromosomes could affect our results, and recommendations for work to focus in on such dynamics in the future in the discussion (full details in answer to question u). Specific questions related to the two chromosomes are address in questions m and u below.

Figure 1: Frequency and distribution of SNPs in Chromosome 1 (a) and Chromosome 2 (b) of VpST3 genomes

More detailed questions:

- a. L26 - We found a VpST3 population (LatAm-VpST3) was introduced > We found a VpST3 population (LatAm-VpST3) that was introduced**

This text has been amended in the abstract.

- b. L59-66; L75-77 – These sentences require some rewriting to read more fluently and make its content clearer to the readers.**

These lines have been amended to read more fluently.

The first sentence now reads: “Before the 1990s, *Vibrio* infections were considered rare, exotic outcomes of foreign travel to tropical Asian waters, apart from particular *V. cholerae* strains that underwent global expansion through pandemic waves (6).” (Lines 60-62)

The second sentence has been rewritten as two separate sentences to aid understanding: “Within months of its initial detection in Asian countries, VpST3 was reported for the first time outside of this endemic region, in the vastly distinct climate of Latin America. This discovery of a VpST3 isolate in February 1996 in Trujillo (11), a city in coastal north-western Peru, was considered an epidemic expansion from the emergence in India.” (Lines 76-79)

- c. L109-110 – Please include the number of new isolates added here as well as the number of publicly available genomes that were used in current analyses.**

The number of isolates (both new and publicly-available) has been added into this sentence in the introduction. It now reads: “A global collection of 280 publicly-available VpST3 genomes, and 32 novel genomic samples from Latin America, were utilized to reconstruct the population structure of

VpST3 globally and quantify the genetic differentiation between samples found in Latin America to those dominant in Asia, and within sub-populations themselves.” (Lines 111-114)

d. L498-511 – Please include the used barcoding approach for the MiSeq sequencing. How was the DNA obtained, what was the QC of the DNA?

We have added greater information in the methodology on how the DNA was obtained as follows: “The strains were retrieved from a –80°C storage freezer, transferred to Luria-Bertani (LB) medium (with 3% NaCl), and incubated at 37°C with shaking at 250 rpm. The genomic DNA was extracted with the DNeasy Blood and Tissue kit (Qiagen, Valencia, CA) from overnight cultures. The DNA concentration was determined using a Qubit double-stranded DNA BR assay kit and a Qubit fluorometer (ThermoFisher Scientific, Waltham, MA), according to manufacturer’s instructions.” (Lines 537-542). The barcoding approach was within the workflow offered by the Nextera XT DNA Library Preparation Kit which uniquely barcodes the samples. We describe this in the text in the methods section- “Sequencing libraries were prepared with the Nextera XT DNA sample preparation kit (Illumina)” (Line 543). The QC of the new raw sequences have been added in Table 2- all have a quality above Q30, and coverage averaging around 100X, so a high quality for these new genomes.

e. L514-535 – Some extended questions on this paragraph:

Which settings were used in the MLST software and how did the authors handle samples that have submission dates instead of actual sampling dates in the field of selection for their metadata.

We have clarified the MLST methodology by listing the housekeeping genes used, and adding further information on the origin of the scheme (PubMLST database) and including a link to the software used (noting default settings were chosen). When identifying publicly-available genomes to use for this study, one of the selection criteria was that the isolate needed to be accompanied by a ‘collection date’, which on NCBI is described as “the time the sample was collected”, to facilitate the spatiotemporal analysis needed. Any samples lacking this metadata were not selected for this study.

We have increased the clarity of both of these decisions in the methodology, which now reads: “Strains were selected from global databases based on two criteria; the samples were accompanied with sufficient metadata (with a sample collection date at a minimum annual resolution, and a location attribute with a minimum country resolution). The sequence type of each genome was obtained using multi-locus sequence typing based on seven housekeeping gene loci across both chromosomes (*dnaE*, *gyrB*, *recA*, *dtbS*, *pntA*, *pyrC* and *tnaA*) from the *V. parahaemolyticus* scheme in the PubMLST database (32) using default settings in MLST v2.11 (<https://github.com/tseemann/mlst>).” (Lines 550-557).

f. Where the original manuscripts consulted for additional verification which is important for any eco-spatio-temporal genomic analyses?

The original manuscripts associated with the publicly-deposited genomes used were consulted as part of an extensive literature review at the initial data collection stage of the study, including to identify further spatiotemporal metadata that was not necessarily included in the database entry.

g. Which tools (i.e., software, versions, settings) were used to assemble the genomes?

The genomes were assembled within the Bactopia v2.0.2 pipeline. While this was mentioned in the text, we acknowledge we could have provided more detail so have now reported the specific assembler this pipeline uses and included a link to the software. The text now reads: “The raw sequence data was processed, validated, quality-filtered, and annotated using the Bactopia v2.0.2

pipeline with default parameters (40). Reads that failed to pass length or quality requirements in the quality check module of the Bactopia v2.0.2 pipeline were filtered out and excluded from downstream analyses. This pipeline then assembled the sequences with default parameters in Shovill v1.1.0 (<https://github.com/tseemann/shovill>), a standard de novo genome assembler for Illumina paired-end reads, with assembly quality statistics generated using QUASt v4.6.0 (41).” (Lines 560-566)

h. Why was a QC filtering of Q10 applied for Illumina MiSeq reads while the median raw read accuracy aims at Q30 reads, which is also intrinsic for QC reporting within the MiSeq device?

All of the new sequences have a quality above Q30, which we have made clearer within the new table (Table 2). This filtering mentioned did not apply to the MiSeq reads, and was instead referring to the quality control stage within the Bactopia pipeline, when all the reads (for both these samples and those from public databases) were later processed. The text in the ‘Genomic data and processing’ sub-heading has now been amended to limit confusion: “Reads that failed to pass length or quality requirements in the quality check module of the Bactopia v2.0.2 pipeline were filtered out and excluded from downstream analyses.” (Lines 561-563)

i. Also, reads of 1kb seems not to be a filtering at the level of raw reads but rather at the level of contigs here?

Similarly, the 1kb read length was incorrectly referred to in regards to the Bactopia pipeline, and not the filtering of the novel raw reads. The read length of the raw reads is now clearer in the new Table 2.

j. Which QC was performed on the the final genome assemblies (i.e., genome completeness/contamination)? Why was no assembly QC implemented in the choice of genomes from the databases?

Thank for your comment. We realise it was not originally clear enough, but we took the raw sequence data (in fastQ reads format) for all the isolates used and generated assemblies ourselves (these all underwent the same analysis pathway to ensure no errors from alternative techniques were introduced), rather than downloading the assemblies available in public databases. We have made this clearer in the text by adding the following sentence, “We downloaded the raw sequence data (in fastQ format) for each VpST3 sequence, which were then run through the same analysis pipeline to ensure consistency.” (Lines 558-560)

We have specified that reads that failed to pass quality requirements were excluded from downstream analyses, and not assembled, however we have added information on the quality statistics additionally generated for each assembly using QUASt- all of these were retained for further analysis, with the previous quality-control step related to the reads providing the exclusion criteria. To make this clear, we reordered the methodology so the assembly method is now read chronologically after the first quality-control exclusion. The text now reads: “The raw sequence data was processed, validated, quality-filtered, and annotated using the Bactopia v2.0.2 pipeline with default parameters (40). Reads that failed to pass length or quality requirements in the quality check module of the Bactopia v2.0.2 pipeline were filtered out and excluded from downstream analyses. This pipeline then assembled the sequences with default parameters in Shovill v1.1.0 (<https://github.com/tseemann/shovill>), a standard de novo genome assembler for Illumina paired-end reads, with assembly quality statistics generated using QUASt v4.6.0 (41).” (Lines 560-566). We therefore did need to quality check the assemblies available in public databases, as we knew the

stage of the pipeline dealing with the raw sequences themselves would automate the removal of any below the set quality requirements.

k. The authors write that annotation was performed within Bactopia, though also with PGAP (L503)?

Thank you to the reviewer for pointing out this inconsistency. While the novel samples were annotated with PGAP at the time of sequencing, the entire VpST3 genome collection was annotated at the same time in the Bactopia pipeline for this study and these were the annotations used in downstream analyses, therefore the text in L503 has been removed as it was irrelevant to this study.

l. How does Gubbins handle highly contiguous sequencing data and how does this impact any further data analyses?

Gubbins' algorithm is based on sliding windows, a spatial scanning statistic, that search for differences in inferred substitution rates, therefore if a window does not fit that substitution rate it is considered to be recombination (and is removed in the resulting non-recombining alignment). The program works on a core genome alignment, and then infers the substitution rate. In consequence, we do not anticipate any differences in highly contiguous genomic data or in a fragmented genome when the program applies these sliding windows. The size for the sliding windows is optional and can be adjusted based on your data (for example very small genomes might require a smaller window size, and those with large recombinations a larger window)- ours was set at a minimum of 100 and maximum of 1000.

m. How do these tools handle bacterial genomes that have multiple chromosomes, as *Vibrio* sp. has two chromosomes that might have different evolutionary pressure? It seems this is not considered in current manuscript, which seems an important aspect when working with *Vibrio* sp.

This is still a very open question; we found very little studies that are exploring the role of different evolutionary pressure on these chromosomes beyond speculation. However, assembly tools (such as Bactopia) are designed to handle multiple chromosomes found in certain bacteria genomes such as *Vibrio*. They ensure that both chromosomes are accurately represented in the final genome sequence. It would be possible to run the tools separately on a separate alignment for each chromosome, but this would give different results that might contradict, so we used the whole genome to get the whole picture.

n. What is the potential impact of using different tools to get to SNPs and why can the same software not be used for both (cf. Parsnp Versus snp-sites and SnpEff). How does this relate to outputs from software tools such as Snippy (<https://github.com/tseemann/snippy>) and roary (<https://sanger-pathogens.github.io/Roary/>), which present more holistic approaches.

Parsnp was used to identify core SNPs to create the core genome alignment due to its ability to rapidly and simultaneously analyse a large number of sequences (such as in the case of our large VpST3 collection), for computational efficiency, as it has been found to be “orders of magnitude faster” than other whole-genome alignment and SNP typing tools (Treangen et al., 2014), which is why we selected this tool over snippy. We then separately used snp-sites to convert these SNP outputs into a variant call format (snp-sites benefits from being able to output multiple formats for a range of downstream analysis), which was needed for the input into the snpeff analysis which predicted the functional effects each SNP was having on genes. We do not anticipate this conversion

would have a large impact, however note that the pipeline could be made more streamlined in the future by only using one of these tools, and exploring different output format conversions. One of the tools suggested by the reviewer, Roary, was already used within the analysis to reconstruct the pangenome and look beyond the core genome, adding extra elements of analysis in terms of exploring accessory gene presence.

o. Was normal bootstrapping or ultrafast bootstrapping applied for the IQtree command? How many valid sites remained after processing and how was the redundancy? Was any down sampling required or performed?

Normal bootstrapping was applied for 1000 replicates to assess phylogeny robustness under a transversion model with equal base frequencies and ascertainment bias correction (ASC) models (identified as best fit using a Bayesian Information Criterion analysis). In terms of valid sites and redundancy, out of 2334 columns, our alignment had 818 parsimony-informative sites. No downsampling was applied. We would like to note however that IQtree was used to make a maximum likelihood phylogeny for an initial exploration into the phylogenetic context of the collection, and to identify the need for a robust phylogeographic approach taking into account the geography of the collection (which was then carried out using BEAST2), and so, while we did a Bayesian Information Criterion analysis to identify the best-fit model, a full sensitivity analyses was not conducted for the IQtree analysis.

We have added this further information into the text as follows: “We constructed a maximum-likelihood phylogenetic tree from the non-recombining SNPs in IQ-TREE v2.2.3 (47), with 1000 normal bootstrap replicates and a TVMe+ASC substitution model (a transversion model with equal base frequencies and ascertainment bias correction, identified as the best-fit model according to a Bayesian Information Criterion analysis), rooted to the pre-pandemic strain, in which 818 sites of the 2334 were found to be parsimony-informative.”(Lines 575-580)

p. L539-555 – How did the authors confirm the choice of their molecular clock model and coalescent? Was any model selection approach, using MLE performed to reassure the right models were applied (https://beast.community/model_selection_1) and (https://beast.community/model_selection_2). Without proper model selection the authors cannot guarantee the best fit model was applied for current analyses.

Thank you for raising this point as we agree we did not sufficiently explain the choices behind our BEAST2 model setup. Here we provide details on the choices of molecular clock, coalescent model and substitution model to ensure these were the best fit, and how we have added this information into the manuscript.

Firstly, our choice of molecular clock was based on previous research already conducted by our team which included extensive path sampling and stepping-stone sampling on the same organism to identify the best molecular clock to be an uncorrelated lognormal molecular clock (Martinez-Urtaza et al., 2017, mBio). Moreover, the model was run for 250,000,000 states and confirmed its convergence of the MCMC outputs (ESS values >200). We have added this decision into the text which now reads as “A relaxed log normal molecular clock model was selected based on previous path sampling and stepping-stone sampling analysis that identified this molecular clock model as the best fit for *V. parahaemolyticus* (4)”. (Lines 591-593)

In addition to this precedent, there are biological reasons for the choice of these models. First of all, our model operates within the coalescent framework, which is well-suited for genealogical inference and demographic history reconstruction. It assumes that the organisms being analyzed derive from a

common ancestor, which is suitable for most pathogen studies as they often originate from a single strain or a few closely related strains, as is the case of Vp-ST3. Additionally the structured coalescent model was chosen for its functionality that facilitated the spatiotemporal analysis necessary, by allowing for regions to have distinct coalescent rates and migration between regions. The assumption of the uncorrelated lognormal molecular clock allows for rate variation among lineages. This assumption is biologically realistic, as different lineages often evolve at different rates due to varying selective pressures, generation times, and other factors. This model accommodates the possibility that evolutionary rates can vary independently across the phylogeny. Therefore, with this assumption, our analysis considers the possibility that each genome of Vp has its particular evolutionary rate.

Lastly, BModelTest was implemented within BEAST2 to identify the nucleotide substitution model with the strongest Bayesian posterior probability support, which informed our choice of a general time reversible substitution model. This supported previous studies that have also used this model, or found it to be the best fit, for *V. parahaemolyticus*. This extra detail and choice has now been added into the text: “The model used a GTR (general time reversible) substitution model with a substitution rate prior (ucl.d.mean) set to a normal distribution, as used previously for *V. parahaemolyticus* (31) and identified here as having the strongest Bayesian posterior probability support using the BModelTest model comparison implementation (50).” (Lines 593-596)

Initial choices of other parameters and priors were initially informed by previous literature working with similar clonal *Vibrio parahaemolyticus* genomes, and then sensitivity analyses to refine based on testing for convergence within Beauti, which we mention briefly: “...and sensitivity analyses for convergence was conducted to set remaining parameters and priors.” (Lines 597-598).

We hope by clarifying our decisions, you now agree our choices were the best fit model for our analyses.

q. L559-579 – What was the burn-in percentage that was used? Were any independent runs run and combined (see comment above). The latter is often recommended as it increases the power of probabilities as individual/independent runs are run with different random number seed that increases robustness when run over different runs and combined afterwards using LogCombiner for example.

The burn-in percentage used to both check convergence and within TreeAnnotator to generate the Maximum Clade Credibility Tree was 10%, which is a recommended percentage by the BEAST2 developer team to allow Markov Chain time to reach equilibrium, and a visual inspection of the convergence on Beauti confirmed this burn-in percentage was appropriate.

We thank the reviewer for their suggestion to use LogCombiner of independent runs, and agree this may have saved us time by reaching convergence faster from different starting point seeds, however, confirm that our single run of 250000000 steps spent enough time in equilibrium to provide Effective Sample Size (ESS- a quality-measure of the resulting sample sequence) values for all parameters over 200 (BEAST2 recommendations are that parameters of interest exceed at least 100). We had the computational power to run our model for this many steps to reach convergence, however if we had struggled to achieve this convergence and mixing, then we agree LogCombiner would have allowed shorter, less-computational runs to be combined to achieve similar sufficient mixing.

- r. **L156-163 – In this section it is important to get more insights in the actual probabilities to guide the readers in the confidence of the drawn trees and maps. What is the support for the spread to China and other countries as indicated in Figure 2C?**

We had previously included posterior support for the main branches in the phylogenetic tree (delineating each group) in Figure 2- we had considered adding the posterior support to all branches on Figure 2 but decided this would make the figure unclear and be difficult to read. Instead, we have now added in-text clarifications of the posterior support for the key routes discussed, to provide the reader with more insight into the probabilities of these spreads. For example, “The VpST3 collection then branches off into two groups, around 1963-4 (95% highest posterior density of 1940-1982), into LatAm-VpST3 (posterior support of 0.837) and the Asian-dominant Cluster (posterior support of 0.919), and the latter then diverges again, around 1995, with the Modern Cluster emerging (posterior support of 0.906).” (Lines 141-144). We have also added in-text reporting of posterior probability when exploring the origins of LatAm-VpST3 (“LatAm-VpST3 was estimated to diverge from the rest of the VpST3 collection around 1963-4 (95% highest posterior density of 1940-1982), arriving in Latin America, from a notably long branch, with an MRCA in Latin America estimated around 1984 (95% highest posterior density of 1978-1994), a posterior probability of 0.837, and a 0.869 probability the location of this branch was Latin America.” Lines 154-158) and its spread into new regions (“One branch resulted in lineages that became established in Latin America from 1997-2014 (posterior support of 0.991) and was introduced regionally to Chile in two separate introductions (posterior support of 0.972), and then trans-continently to China and the USA (Figure 2c) with a posterior support of 0.853 and 0.899 on these branches respectively.” Lines 165-169).

- s. **L165-174 – This might also be biased by the used coalescent, which is the Bayesian skyline here. How did the authors test for the proper choice of coalescent to fit the population here? What about the Bayesian Skygrid coalescent model with Hamiltonian Monte Carlo sampling? This allows a more “flexible” fit of the available data to represent the population dynamics. This approach also allows to specify cut-off date and population sizes within the MCMC sampling procedure.**

In our previous study (Martinez-Urtaza et al., 2017, mBio), different demographics (constant population size, exponentially growing population GMRF and Bayesian skyline plot) and molecular clock (strict, random and uncorrelated lognormal) models and finally, a Bayesian skyline demographic model and uncorrelated lognormal molecular clock were finally selected as the best demographic model for this dataset according to path sampling (PS)/stepping-stone sampling (SS) values, selecting it above a Bayesian Skygrid. We also confirm that the ESS (effective sample size) values for this model all exceeded 200, which can be considered a good model fit with sufficient convergence in BEAST2 analysis.

From a biological and evolutionary point of view, the Bayesian skyline model allows for changes in population size over time without assuming a specific demographic model. This flexibility is crucial studying populations like *V. parahaemolyticus*, where, for example, we found some population particularities within lineages like ST36. Unlike parametric models that require the specification of a functional form for population size changes, the Bayesian skyline approach is non-parametric. This means it can capture sudden demographic events, such as bottlenecks or expansions, which we interestingly found in LatAm-VpST3 population. However, we thank the reviewer for their recommendation of Bayesian Skygrid as we understand them to be a flexible approach for other datasets.

- t. L222-228 – What is the confidence on these recombination rates (cf. comment in M&M section)? What is the impact of gene deletions/inserts besides SNPs? Applying different software (e.g., snippy) would also report these indels for functional assessment as this is the “focus” of this section.**

To our knowledge, the per-branch statistics reported by Gubbins (including recombination rates) are not accompanied by confidence or uncertainty values. Accuracy could be tested by comparing recombination rates identified through alternative tools, however Gubbins was selected here as it has been used previously to remove recombining regions for a range of *Vibrio*-specific analysis in previous works and recent studies (Martinez-Urtaza et al., 2017; Janecko et al., 2021; Bhandari et al., 2023). We focused on SNPs rather than indels for multiple reasons; firstly indels are not currently natively supported as input for BEAST 2 analyses; secondly SNPs are an informative and consistent marker for mapping population genetics, and they have more predictable, functional effects (compared to indels that cause frameshifts whose effect is very difficult to predict). We focused our analysis on SNPs as they are frequent and stable to gain insight into the evolutionary history behind the emergence of the particular Latin American group, like most studies exploring the evolutionary mechanisms of *Vibrio* bacteria (Martinez-Urtaza et al., 2017).

- u. It would be of high value to differentiate genes based on their coding chromosome to see if one or the other chromosome has a different evolutionary pressure or not. I believe implementation of this, would shed interesting depth and (new) insights to the understanding in current manuscript. Even performing some BEAST analyses on the separated chromosomes would improve the manuscript.**

We have included in-text references as to the chromosome location of genes such as the fact the seven housekeeping genes used to obtain the sequence type of each genome (*dnaE*, *gyrB*, *recA*, *dtdS*, *pntA*, *pyrC* and *tnaA*) were spread across both chromosomes and adding in references to chromosome location for genes of interest: e.g. “was predicted to have a non-synonymous effect on *mnmE*- a gene located on chromosome 1 and essential for growth” (Lines 389-390), “Four genes located in chromosome 1 and involved in bacterial putrescine pathways (*aguA*, *aguB*, *puuR* and an unnamed gamma-glutamyl-gamma-aminobutyrate hydrolase family protein)” (Lines 438-439), “we found a significant difference between the presence of a sodium-proton antiporter accessory gene (located on chromosome 1) in LatAm-VpST3, where it was always present” (Lines 411-412). Additionally, Table 1 has been appended with this information, with an addition to the first column heading specifying the chromosome the gene was located on. This made it clear to us that most of the genes we focused on were within Chromosome 1 which is the larger of the two chromosomes, so potentially expected. A brief discussion of how this provides insight into different evolutionary pressured has been added into the discussion, “While exploring the specific evolutionary role of each chromosome was outside the scope of this study, most of the accessory genes that emerged as specific to the LatAm-VpST3 population, or associated with environmental variables, were located on the first chromosome, with the exception of a FAD-dependent oxidoreductase gene. This chromosome is the largest of the two, and impacts of mutation and recombination have previously been found to be higher in this chromosome for *V. parahaemolyticus* ST36 (4), however recent genomic insights suggest the second, smaller chromosome may be involved in adaptation to different ecological niches (29, 30). While this was not identified here, specific analysis for each of the two chromosomes (with specific parameters set for each chromosome) would provide further insights into its role in facilitating the expansion of VpST3 in distinct marine environments.” (Lines 459-468).

We believe that the additions to the methodology and discussion based on Reviewer 1's comments have enhanced the clarity of the decisions taken, and has emphasized the insights provided by our findings.

Reviewer #2

This research using a global collection of clinical and environmental VpST3 isolates, based on Vp genome evolutionary analysis, climate associated gene function analysis and oceanic climate changing data found out Vp evolutionary mechanisms driving global expansions of pathogen strains. The finds reshaped the current understanding of the global expansion of the V. parahaemolyticus clone and help us understand function of factors like climate, gene mutation in the pathogen expansion. However, we have a couple of comments to address

We would like to thank the reviewer for their positive feedback and constructive suggestions to improve the manuscript. We have addressed the comments raised, clarifying the spatial distribution of samples to highlight our representation of Latin American countries, including the posterior probability values to demonstrate the support for the conclusions, adding a new table to provide further genomic information on the novel strains, and drawing further attention to the specific environmental tolerances and responses of *Vibrio* bacteria to even the smallest changes in their environment.

1. Only 32 VP strains from Latin, weather these strains well represent the Latin VP? and we want the author describe all the VP outbreaks cases in Latin countries, and demonstrate these 32 strains can represent the Latin VP ST3. All the 32 strains were collected from 1996-2005, it may be hard to collect the strains from decades ago, but why not enroll the strains after 2005, no outbreak cases after 2005?

We have used all the strains available, which included 71 from Latin America, ranging over a time period of 1996-2019 (see Table S1 in supplementary material). The 32 samples referenced in the study refer to the novel samples specifically added to global collections within this study. We have clarified this in the text by indicating how many of each sample were found in each continent: "The trimmed VpST3 collection contained 312 isolates in total from 19 countries within 3 continents (163 from Asia, 71 from South America and 78 from North America)" (Lines 566-567).

In this way, we have used all the *Vibrio parahaemolyticus* strains available to us from Latin America with sufficient quality and metadata. The majority of these strains were of clinical origin, meaning they were reported in health settings where the patient was likely reporting symptoms, which allows us to link this specifically to infections. *V. parahaemolyticus* infections are commonly underreported, however the Peruvian authorities have been taking genomic samples for any cholera-like symptoms since the 1991 cholera epidemic, allowing the concurrent identification of *V. parahaemolyticus*, so this genomic surveillance gives our only possible indication of the cases in Latin American countries. While genomic data can never be fully representative, we believe this is a sufficient representation of *V. parahaemolyticus* in Latin America, relative to the even higher under-reporting found elsewhere.

2. Line 160-163 "LatAm-VpST3 itself (Figure 2b) originates from a single ancestor in Peru, from which most branches bottleneck with limited long-term establishment success, while another results in lineages that became established in Latin America (from 1997-2014) as well as being introduced regionally to Chile (in two separate introductions), and then trans-continently to China and the USA (Figure 2c)". The author tried to clarify the ST3 dissemination routes among Latin-China-USA. But we think the dissemination routes may not be a single way, because the infection sources and the genetic differentiation strains are not occurred in an only unique place. Like the new detected serotype O10:K4(ST3) we firstly reported in Guangxi, south China, but quickly, the north of China detected the O10:K4 as well. It is not likely transmit from south to north, it more likely

that the O10:K4 exist in China but the identified time is different. Besides, in this study, lacking information of the VP single ancestor strains from Peru, and from the Fig2 the Latin American, North American and China ST3 strains distribute in the three clusters, indicates the infectious sources complicates, so the author got the conclusion that the VP single ancestor in Peru then transmit to Chile, China and USA need more supports.

We agree with the reviewer that it is difficult to conclude the LatAm-VpST3 group emerged from a single introduction in Peru, particularly due to the long branch leading to this group, providing no evidence of a common ancestor or any intermediate variants for 10 years, which means multiple introductions were possible. We point out the limitations of this branch, however our data on population diversity indicates (based on expected bacterial population structures) that only one of these ancestors prevailed in terms of surviving and became successful enough to establish to the point these populations were causing infections in humans, as all the human infections were linked to this single variant. This can be found in the discussion:

“While the absence of intermediary ancestral states on the long evolutionary branch to LatAm-VpST3 could infer missing data related to introductions or evolution, the within-population diversity of LatAm-VpST3 was initially low, confirming no evidence for multiple introductions, with VpST3 instead functioning as a background population before the reported emergence in the late 1990s.” (Lines 349-353).

V. parahaemolyticus particularly is not likely to be detected until there are large outbreaks with a large signal. Indeed, this variant was only detected in Latin America because of routine surveillance that was implemented after the *V. cholerae* 1991 epidemic in Peru. We therefore have to make interpretations of results from methods which are intrinsically based on the information (genomes) that we have.

3. The ocean temperature and alkalinity (pH 8.06 and 8.0) variation between Peru and China are small. Do these small differences in environment cause genetic mutations or require genetic mutations to adapt?

In terms of temperature variation, multiple studies have explored *Vibrio* responses to temperature variation, with the conclusion that any increase in environmental temperature (whether it is large or small, short or long-term) significantly effects concentrations of *Vibrio* in water (Froelich and Daines, 2020). Here, one of the notable differences in SST between Peru and China is that Peru has a generally lower SST but this increases significantly during El Niño events (see Supplementary Figure S7). Here we see the effect of El Niño events that bring tropical conditions that may facilitate *Vibrio* populations, however when conditions are restored *Vibrio* need to survive in these non-El Niño periods- these extremes and high-range variation do not occur in other parts of the whole, including China, which the supplementary figures aim to exemplify. Our finding therefore of multiple SNPs associated with positive SST anomalies suggests this ENSO variation was a specific environmental condition that LatAm-VpST3 was exposed to, potentially involved in its specific evolution in this area. We have added a sentence into the discussion to draw attention to the importance of this SST variation: “Specifically, the SNPs associated with positive SST anomalies highlight the potential role of El Niño events in the evolutionary pathway of LatAm-VpST3- with innovation resulting from the increased SSTs and more tropical conditions present during these events, an environmental variation not experienced to the same extent by populations evolving in Asia.”. (Lines 403-407).

We also note that these pH values (8.06 and 8.08) represent an average over the entire coastal area (for Peru and China respectively) for the entire time period, which is averaging across immense areas and so at this point there is very little variation, however we have attempted to visualise in Supplementary Figure S9 the more notable differences between these- specifically the distinct seasonal variation in pH, and the interannual trends that highlight how Peru has a consistently lower pH value over the time period.

Vibrio bacteria can respond very rapidly and opportunistically to even the smallest changes in environmental conditions due to their rapid replication. These changes can introduce selective pressure, which results in genetic mutations that increase survival in the new conditions to be positively selected for, or encourage the horizontal transfer of beneficial accessory genes, facilitating adaptation over generations to these specific environmental conditions. The identification of a gene in LatAm-VpST3 promoting survival in acidic conditions under positive selection suggests the variation could have been sufficient enough that this gene increased environmental fitness in these conditions.

4.The authors using a pangenome-wide association study tool to find the sea surface temperature (SST) relative genes, and tried to elucidate the Latin-ST3 adaptation mechanism. It found out some adaptation genes may associate with the SST, but lacking the experiment that proved these genes have the function in adaptation. Add a gene function experiment can well validate these genes function and confirm the speculation.

We agree that gene function experiments are critical for validating and providing further insight to the environmental associations we discuss here. Unfortunately, we are not currently able to conduct such experiments as we only have access to sequence data currently, and not live strains. However, this is something we would plan to do in the future- as such, we have added this recommendation into the discussion as follows: “In the future, live strains should be collected to facilitate gene function experiments which are necessary to confirm the associations identified here in real-time.” (Lines 432-434).

5.In this study, lack of VP ancestral strains from Latin America and all the VP strains used in this study were collected after 1996. So in the evolution analysis, only based on the SNP analysis, it is hard to get the conclusion that the ST3 pathogen introduction in Latin America in 1985.

Reconstruction of ancestral states in bacteria, where there is no fossil record, have to be based on inference from genomic data. From the methodological perspective, BEAST2 was used for its ability to look back in time and make inferences on the timing of evolutionary stages, based on statistically inferring the most-recent common ancestor of genomic samples based on genetic differentiation and estimated mutation rates- allowing us to look back to a time period where we do not have genetic samples. While any inference comes with assumptions and caveats, it is one of the most robust methodologies for such analysis. The posterior probability of this decision has now been added into the text, to quantify the certainty of this temporal estimation- “arriving in Latin America, from a notably long branch, with an MRCA in Latin America estimated around 1984 (95% highest posterior density of 1978-1994), a posterior probability of 0.837, and a 0.869 probability the location of this branch was Latin America.” (Lines 155-158).

Conceptually, in our discussion, we mention how a variant of *Vibrio cholerae* (El Tor) was also predicted to have been introduced in Latin America around this date, far before the detection of

strains related to the 1991 epidemic (Donman et al., 2017). This is common for *Vibrio* bacteria due to their ability to exist in a viable but non-culturable state (Wu et al., 2016). This means *Vibrio* populations can lie dormant in the marine environment until optimum environmental conditions arrive which triggers rapid population growth (such conditions as the 1997 intense El Niño event which brought more tropical conditions to the Latin American Pacific coast). Additionally, there would need to be significant population expansion from the point of initial introduction, to get to the point where Vp populations are so abundant and concentrated in the marine environment that humans come into contact with them and present illness significant enough for them to visit a medical setting, for clinical samples to be available. Therefore, it is likely these initial populations exist in the marine environment far before clinical samples would become available for study. We have expanded therefore on our reference to *V. cholerae* El Tor, to explain this process, by adding: "Such temporal lag is expected for *Vibrio* populations; firstly due to the population growth required to result in human transmission to the point of clinical samples being acquired in medical settings, and secondly due to the specific ability of *Vibrio* to lie dormant as viable but nonculturable (VBNC) cells (18) until optimum environmental conditions arrive- in the case of LatAm-VpST3, a strong El Niño event bringing tropical conditions in 1997." (Lines 372-376).

6. Suggesting listing more detailed genetic information of the 32 newly ST3 strains of *Vibrio parahaemolyticus* from Latin. And show more differences between strains prevalent in Asia in 1996.

The metadata for the 32 new strains can be found in Supplementary Table S1, marked in the final column as a new sample. We have also now added further genomic information on the 32 novel VpST3 strains within a new table (Table 2), including detailed information on length, quality and coverage.

Under the sub-heading "Genetic Differentiation of LatAm-VpST3" in the results, we have provided multiple metrics that quantify the difference between LatAm-VpST3 and the rest of the VpST3 collection, including AMOVA and F_{ST} (finding a "6.4% and 5.5% significant ($p=0.001$) genetic differentiation" respectively in Lines 202-203). We also compared two methods that identified LatAm-VpST3 as a significant unique 'cluster' within the collection, using TreeStructure and DAPC (discriminatory analysis of principle components, both supervised and unsupervised), which had a 98.2% agreement. Some of the specific differences identified were 3 SNP mutations unique to LatAm-VpST3 in the core genome (at positions 235, 957 and 1010), and, within the pangenome, 11 accessory genes whose presence was significantly sensitive to LatAm-VpST3 (always present in this group, not necessarily always present outside), and 3 accessory genes whose presence was significantly specific to LatAm-VpST3 (uniquely found in this group, absent in the rest of the collection)- see Table 1 in the manuscript for the specific genes and functions. Such significant ($p<0.05$ findings) support the genetic difference between LatAm-VpST3 and the remaining collection. Figure 4 visualises the results of the DAPC analysis, showing much closer genetic relatedness (and admixture) between the Asian-Dominant Cluster and the Modern Cluster- we would expect to see such closeness with the LatAm-VpST3 group but it is largely separate in the RDA space. We did not specifically report the genetic difference between the early samples (1996-1997), however an F_{ST} of 1.3% genetic differentiation was found between early samples from the LatAm-VpST3 and Asian-dominant group.

7. In this study, some bioinformatics tools were used to calculate and analyse the evolutionary

relationship. Like the strains in Fig2 1a are all collected after 1997, how it can draw the result that event occurred before 1940-1998. Most of the results in this paper based on the evolutionary analysis. The author should fully discuss these tools and models used can derive the correct speculation or results.

We have specified in Question 5 the specifics on how Bayesian models (using BEAST2) provide historical inferences when reconstructing evolutionary pathways. Additionally, we agree we should provide the reader with further information on this, so have added the following sentence in the Materials and Methods section: "VpST3 dynamics were reconstructed using a Bayesian phylogeographic approach in BEAST v2 (43), a package that facilitates the statistical inference of historic population dynamics and evolutionary parameters from molecular sequence data using a Markov chain Monte Carlo (MCMC) algorithm sampling from a posterior distribution." (Lines 585-588).

We have also now highlighted the specific probability assigned to each of these estimated dates from our Bayesian phylogenetic analysis, to give an indication of certainty that is pertinent to subsequent findings, e.g. "The VpST3 collection then branches off into two groups, around 1963-4 (95% highest posterior density of 1940-1982), into LatAm-VpST3 (posterior support of 0.837) and the Asian-dominant Cluster (posterior support of 0.919), and the latter then diverges again, around 1995, with the Modern Cluster emerging (posterior support of 1)." (Lines 141-144).

Reviewer #3

Overall, a well-conceived and executed study.

We thank the reviewer for spending the time to read our manuscript and providing positive and constructive comments. We have added further discussion on the further environmental variables that could influence evolutionary dynamics of VpST3 and the spatiotemporal metadata limitations preventing their inclusion in the current study.

While there may be an association of SNPs with SST, that is not surprising given that is the main variable examined. Caution making too strong of conclusions when other environmental factors (dissolved oxygen, etc.) that may influence genetic change were not examined. This possibility should be acknowledged and appropriately discussed.

We agree that there are a range of environmental factors that can affect *Vibrio* bacteria in the marine environment. Salinity and temperature have been identified as the major drivers of *Vibrio* distribution and abundance (Baker-Austin et al., 2018), leading to use using these as our main environmental proxies, however other environmental variables have secondary roles in these processes, including chemical components (dissolved oxygen and carbon) and biological components (plankton presence) in the marine environment. We agree further environmental variables should be examined in the future, but suggest that some of these more localised dynamics (including dissolved nutrients) would require higher spatiotemporal metadata relating to the location and date of genomic sample collection, to facilitate spatiotemporal analysis. We therefore focused on granular resolution physical variables such as temperature and salinity, that exhibit relatively less regional variation than some of the variables mentioned above, to highlight particular mechanisms by which marine environmental variables (and those most commonly explored in terms of *Vibrio* ecology) are involved in the evolutionary pathways of VpST3. Additionally, in regards to dissolved oxygen specifically, sea surface temperature can modulate oxygen solubility and therefore dissolved oxygen content in marine environments, so the impact of SST on genetic change could be linked to such associations with dissolved oxygen, but this would require higher resolution analysis than is currently possible with the available VpST3 genomes.

We have added this discussion into the text as follows: “Higher spatiotemporal resolution of genomic metadata will enhance our ability to retrospectively reconstruct *Vibrio* population dynamics in the environment and the variables driving these. This includes the future inclusion of further environmental variables associated with evolutionary mechanisms that have more local biochemical dynamics, such as chitin availability from planktonic organisms which can increase environmental DNA uptake (23), or dissolved oxygen which is associated with the presence of virulence factors (24). While our analysis found particular associations with SST and salinity, such variables could be providing proxy information on other components of the dynamic and interconnected marine ecosystem, from which higher-resolution analysis is required to isolate specific environmental associations.” (Lines 423-434).

Reference List

Baker-Austin, C., Oliver, J. D., Alam, M., Ali, A., Waldor, M. K., Qadri, F., & Martinez-Urtaza, J. 2018. *Vibrio* spp. infections. *Nature Reviews Disease Primers*, 4(1), 1-19. doi.org/10.1038/s41572-018-0010-y

Bhandari, M., Rathnayake, I.U., Huygens, F., Nguyen, S., Heron, B. and Jennison, A.V., 2023. Genomic and evolutionary insights into Australian toxigenic *Vibrio cholerae* O1 strains. *Microbiology Spectrum*, 11(1), pp.e03617-22. doi: 10.1128/spectrum.03617-22

Domman et al. 2017. Integrated view of *Vibrio cholerae* in the Americas. *Science* 358, 789-793. DOI:10.1126/science.aao2136

Froelich, B.A. and Daines, D.A. 2020. In hot water: effects of climate change on *Vibrio*–human interactions. *Environ Microbiol*, 22: 4101-4111. <https://doi.org/10.1111/1462-2920.14967>

Janecko, N., Bloomfield, S.J., Palau, R. and Mather, A.E., 2021. Whole genome sequencing reveals great diversity of *Vibrio* spp in prawns at retail. *Microbial Genomics*, 7(9), p.000647.

Kirkup, B.C., Chang, L., Chang, S. et al. 2010. *Vibrio* chromosomes share common history. *BMC Microbiol* 10, 137. <https://doi.org/10.1186/1471-2180-10-137>

Martinez-Urtaza Jvan Aerle R, Abanto M, Haendiges J, Myers RA, Trinanes JBaker-Austin C, Gonzalez-Escalona N. 2017. Genomic Variation and Evolution of *Vibrio parahaemolyticus* ST36 over the Course of a Transcontinental Epidemic Expansion. <https://doi.org/10.1128/mbio.01425-17>

Treangen, T.J., Ondov, B.D., Koren, S. et al. 2014. The Harvest suite for rapid core-genome alignment and visualization of thousands of intraspecific microbial genomes. *Genome Biol* 15, 524. <https://doi.org/10.1186/s13059-014-0524-x>

Wu, B., Liang, W., & Kan, B. 2016. Growth phase, oxygen, temperature, and starvation affect the development of viable but non-culturable state of *Vibrio cholerae*. *Frontiers in microbiology*, 7, 183163.

REVIEWERS' COMMENTS

Reviewer #1 (Remarks to the Author):

No further comments; All my questions have been answered and fulfilled.